# The E2 Marie Kondo and the CTLH E3 ligase clear deposited RNA binding proteins during the maternal-to-zygotic transition

Michael Zavortink[1†‡], Lauren N Rutt[1†], Svetlana Dzitoyeva[1†§],
Jesslyn C Henriksen[1], Chloe Barrington[1], Danielle Y Bilodeau[1], Miranda Wang[2],
Xiao Xiao Lily Chen[2], Olivia S Rissland[1]*

[1]University of Colorado School of Medicine, Aurora, United States; [2]University of Toronto, Toronto, Canada

**Abstract** The maternal-to-zygotic transition (MZT) is a conserved step in animal development, where control is passed from the maternal to the zygotic genome. Although the MZT is typically considered from its impact on the transcriptome, we previously found that three maternally deposited *Drosophila* RNA-binding proteins (ME31B, Trailer Hitch [TRAL], and Cup) are also cleared during the MZT by unknown mechanisms. Here, we show that these proteins are degraded by the ubiquitin-proteasome system. Marie Kondo, an E2 conjugating enzyme, and the E3 CTLH ligase are required for the destruction of ME31B, TRAL, and Cup. Structure modeling of the *Drosophila* CTLH complex suggests that substrate recognition is different than orthologous complexes. Despite occurring hours earlier, egg activation mediates clearance of these proteins through the Pan Gu kinase, which stimulates translation of *Kdo* mRNA. Clearance of the maternal protein dowry thus appears to be a coordinated, but as-yet underappreciated, aspect of the MZT.

*For correspondence:
olivia.rissland@gmail.com

†These authors contributed
equally to this work

Present address: ‡Peter
MacCallum Cancer Centre,
Melbourne, Australia;
§Northwestern University,
Evanston, United States

**Competing interests:** The
authors declare that no
competing interests exist.

**Reviewing editor:** Michael B
Eisen, University of California,
Berkeley, United States

## Introduction

Proper embryogenesis is critical for animal development. Many of the earliest events occur prior to the onset of zygotic transcription, and they are instead directed by maternally deposited proteins and messenger RNAs (mRNAs). During the maternal-to-zygotic transition (MZT), genetic control of developmental events changes from these maternally loaded gene products to newly made zygotic ones (*Vastenhouw et al., 2019*). Thus, the MZT requires both the activation of zygotic transcription and clearance of maternal transcripts. Failure to mediate either of these processes is lethal for the embryo (*Benoit et al., 2009*; *Liang et al., 2008*).

In contrast to our understanding of the transcriptome during the MZT, much less is known about changes in the proteome. Despite the fact that the maternal dowry of proteins plays key roles in embryogenesis, there are only a handful of examples of cleared maternal proteins (*Guven-Ozkan et al., 2008*; *Hara et al., 2017*; *Pesin and Orr-Weaver, 2007*; *Sysoev et al., 2016*; *Wang et al., 2017*; *Yang et al., 2016*). Recently, we found that three RNA-binding proteins (ME31B, Trailer Hitch [TRAL], and Cup) are rapidly degraded during the MZT in *Drosophila melanogaster*, at a time point coinciding with the major wave of zygotic transcription (*Wang et al., 2017*). ME31B, TRAL, and Cup form a complex that blocks translation initiation (*Kinkelin et al., 2012*; *Nakamura et al., 2004*; *Nelson et al., 2004*; *Wilhelm et al., 2003*). All three proteins are required for oogenesis, and they appear to bind and repress thousands of deposited maternal mRNAs (*Keyes and Spradling, 1997*; *Nakamura et al., 2001*; *Tritschler et al., 2008*; *Wang et al., 2017*; *Wilhelm et al., 2003*). The degradation of ME31B, TRAL, and Cup coincides with many of the

**eLife digest** Bestselling author and organizing consultant Marie Kondo has helped people around the world declutter their homes by getting rid of physical items that do not bring them joy. Keeping the crowded environment inside a living cell organized also requires work and involves removing molecules that are no longer needed. A fertilized egg cell, for example, contains molecules from the mother that regulate the initial stages as it develops into an embryo. Later on, the embryo takes control of its own development by destroying these inherited molecules and switches to making its own instead. This process is called the maternal-to-zygotic transition.

The molecules passed from the mother to the egg cell include proteins and messenger RNAs (molecules that include the coded instructions to make new proteins). Previous research has begun to reveal how the embryo destroys the mRNAs it inherits from its mother and how it starts to make its own. Yet almost nothing is known about how an embryo gets rid of its mother's proteins. To address this question, Zavortink, Rutt, Dzitoyeva et al. used an approach known as an RNA interference screen to identify factors required to destroy three maternal proteins in fruit fly embryos.

The experiments helped identify one enzyme that worked together with another larger enzyme complex to destroy the maternal proteins. This enzyme belongs to a class of enzymes known as ubiquitin-conjugating enzymes (or E2 enzymes) and it was given the name "Kdo", short for "Marie Kondo". Further experiments showed that the mRNAs that code for the Kdo enzyme were present in unfertilized eggs, but in a repressed state that prevented the eggs from making the enzyme. Once an egg started to develop into an embryo, these mRNAs became active and the embryo started to make Kdo enzymes. This led to the three maternal proteins being destroyed during the maternal-to-zygotic transition.

These findings reveal a new pathway that regulates the destruction of maternal proteins as the embryo develops. The next challenge will be identifying other maternal proteins that do not "spark joy" and understanding the role their destruction plays in the earliest events of embryonic development.

hallmarks of the MZT, but explorations into this issue have been hindered by a lack of understanding of how their destruction is controlled.

We previously made an intriguing observation that genetically linked the clearance of ME31B, TRAL, and Cup, to the Pan Gu (PNG) kinase (*Wang et al., 2017*). Composed of three subunits (PNG, Giant Nuclei [GNU], and Plutonium [PLU]), the PNG kinase is central to the oocyte-to-egg transition and mediates key aspects of embryogenesis, including resumption of the cell cycle, zygotic transcription, and maternal mRNA clearance (*Elfring et al., 1997*; *Tadros et al., 2007*; *Vardy and Orr-Weaver, 2007*). Unlike many animals, the oocyte-to-egg transition in *Drosophila* does not require fertilization but is instead triggered by egg activation (*Doane, 1960*; *Heifetz et al., 2001*; *Horner and Wolfner, 2008a*). Here, the PNG kinase is activated by mechanical stress as the oocyte passes through the oviduct, and then phosphorylation and degradation of the GNU subunit quickly inactivates the kinase, restricting its activity to the first half hour after egg activation (*Hara et al., 2017*). One way that PNG mediates the oocyte-to-embryo transition is by rewiring post-transcriptional gene regulation (*Eichhorn et al., 2016*; *Kronja et al., 2014*). Possibly by phosphorylating key RNA-binding proteins such as Pumilio, PNG activity leads to changes in the poly(A)-tail length and translation of thousands of transcripts during egg activation (*Hara et al., 2018*). Importantly, two targets induced by PNG activity are the pioneer transcription factor Zelda, which is responsible for initial zygotic transcription, and the RNA-binding protein Smaug, which is responsible for clearance of many maternal transcripts (*Benoit et al., 2009*; *Eichhorn et al., 2016*; *Liang et al., 2008*; *Tadros et al., 2007*; *Vardy and Orr-Weaver, 2007*). The PNG kinase also phosphorylates ME31B, Cup, and TRAL (*Hara et al., 2018*), but it is unclear what effect phosphorylation has on these proteins. One possibility has been that PNG phosphorylation could lead to the degradation of ME31B, TRAL, and Cup, but this model has been thus far unexplored.

The ubiquitin-proteasome system is a major protein degradation pathway. Here, a series of ubiquitin activating enzymes, conjugating enzymes, and ligases (E1, E2, and E3, respectively) lead to the

post-translational addition of a polyubiquitin chain on a target protein, which then serves as a molecular beacon for degradation by the proteasome. E3 ligases are typically thought to recognize target proteins, while E2 conjugating enzymes provide the activated ubiquitin and in turn recognize the E3 ligase (*Komander and Rape, 2012*). There are hundreds of different E3 ligases and 29 annotated E2 conjugating enzymes in *Drosophila* (*Du et al., 2011*), but most of the client substrates are unknown, and few have been implicated in the MZT.

Given the key roles of ME31B, Cup, and TRAL in oogenesis and embryogenesis, we wanted to understand the mechanisms controlling their degradation. In particular, we sought to answer how PNG activity at egg activation leads to the degradation of these three RNA-binding proteins several hours later, and how their degradation is coordinated with other elements of the MZT, including zygotic transcription and maternal mRNA clearance. To answer these questions, we performed a selective RNAi screen in *Drosophila*, and identified the E2 conjugating enzyme as UBC-E2H/ Marie Kondo and the E3 ligase as the CTLH complex. Interestingly, structural models based on the *S. cerevisiae* complex (*Qiao et al., 2020*) suggest that the *Drosophila* version is organized differently than its orthologous complexes. The CTLH complex recognized and bound ME31B and Cup even in the absence of PNG activity, strongly suggesting that phosphorylation is not required for the destruction of these proteins. In contrast, *Kdo* mRNA is translationally upregulated by more than 20-fold upon egg activation in a PNG-dependent manner. Thus, egg activation through PNG mediates translation upregulation of *Kdo* and so leads to ME31B, Cup, and TRAL destruction.

## Results

### PNG kinase activity at egg activation triggers destruction of ME31B

We previously demonstrated by western blotting that ME31B, TRAL, and Cup were degraded 2–3 hr after egg laying (*Wang et al., 2017*). To understand the mechanisms underlying degradation of these RNA-binding proteins, we decided to establish a fluorescence-based assay so that we could follow ME31B degradation in living embryos. To do so, we took advantage of an ME31B-GFP trap line where the fusion protein is expressed from the endogenous locus (*Buszczak et al., 2007*); we have previously shown that ME31B-GFP recapitulates the dynamics of the wild-type protein (*Wang et al., 2017*). Consistent with western blotting, the GFP signal in control ($png^{50}$/FM7) embryos robustly decreased from 2 to 3 hr after egg laying (*Figure 1A,B*). In contrast, the GFP signal remained constant through this time period in $png^{50}$/$png^{50}$ embryos (hereafter referred to as $png^{50}$), consistent with our previous study (*Wang et al., 2017*). Note that less heterogeneity in fluorescence was also observed in the mutant embryos; this observation likely stems from the fact that in wild-type embryos some ME31B-GFP degradation occurs during embryo collection and staging, but almost none occurs in $png^{50}$ embryos. Together, these results confirm that the differences in ME31B-GFP dynamics are observable by microscopy and that the degradation of ME31B-GFP requires PNG.

To test the importance of other genes for ME31B degradation, we combined the ME31B-GFP allele with a GAL4-UAS system, where GAL4 was under the control of the matα-tubulin promoter and so is specifically expressed during oogenesis; this system enabled us to induce the expression of dsRNA during oogenesis and monitor the requirement of various genes for ME31B-GFP degradation. Given that $png^{50}$ embryos did not degrade ME31B-GFP, we first investigated whether the other two components of the PNG kinase, GNU and PLU, were required. When either GNU or PLU were knocked down (*Figure 1C,D*), ME31B-GFP was again stabilized, thus confirming that its destruction requires the full PNG kinase.

We next asked whether the degradation of ME31B-GFP required fertilization by following ME31B-GFP levels with fluorescence microscopy in unfertilized eggs (*Figure 1E*). In contrast to our results in $png^{50}$ embryos, ME31B-GFP was still unstable in the activated, but unfertilized, eggs. This result is consistent with numerous studies demonstrating that the major events pre-MZT in *Drosophila* require egg activation (primarily through the PNG kinase), but not fertilization (*Fenger et al., 2000*; *Horner and Wolfner, 2008b*; *Tadros et al., 2007*; *Tadros et al., 2003*; *Vardy and Orr-Weaver, 2007*). Taken together, we conclude that degradation of ME31B is triggered by egg activation through PNG activity.

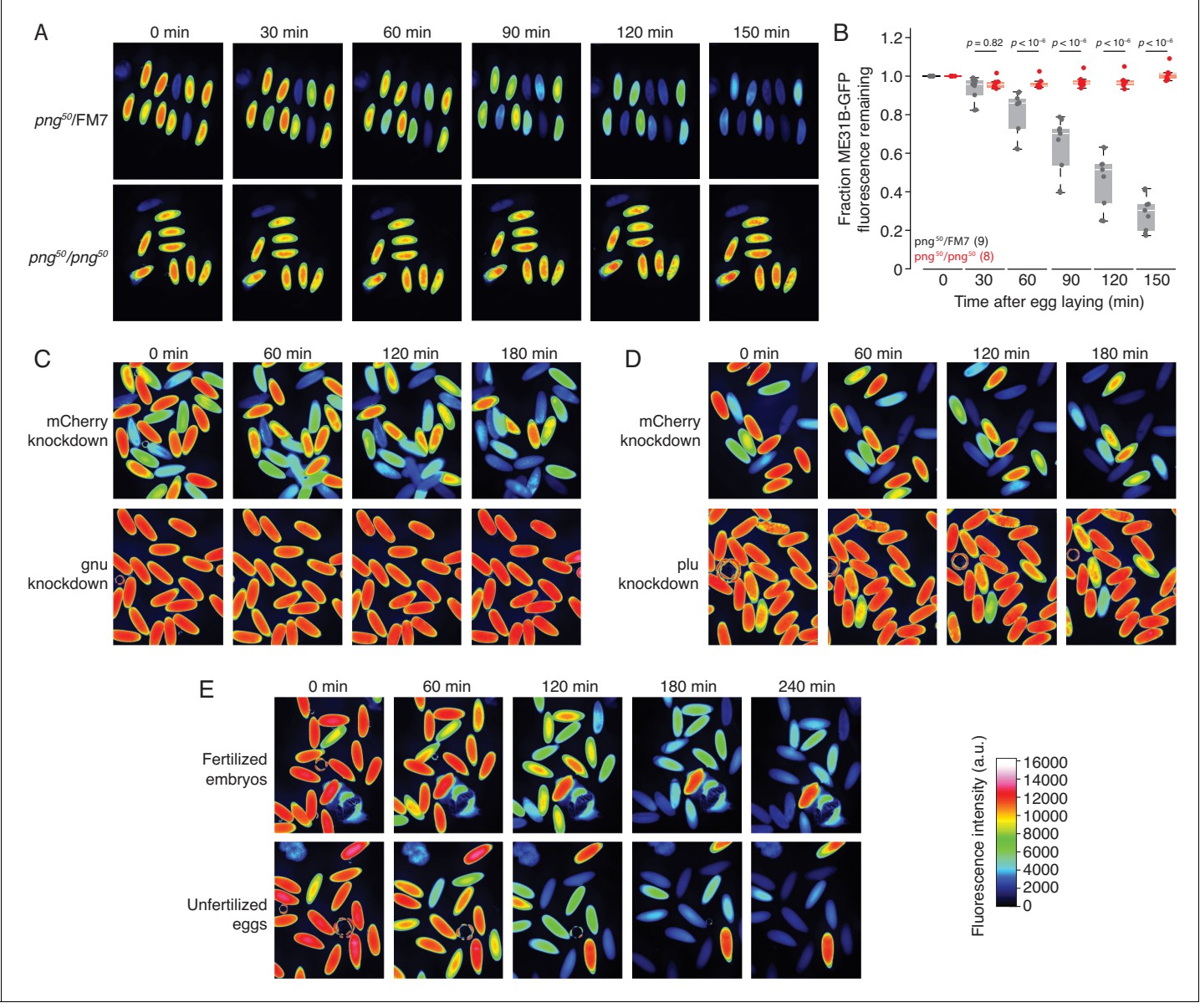

**Figure 1.** The PNG kinase, but not fertilization, is required for ME31B degradation. (**A**) PNG is required for the destruction of ME31B. Embryos from female flies of the indicated genotypes (that also contained the *ME31B-GFP* allele) were visualized at various time points after egg laying. Fluorescent images are false-colored so that more intense fluorescence is indicated by hotter colors; fluorescence scale is shown at bottom of figure. (**B**) ME31B-GFP is stabilized in *png* mutants. ME31B-GFP fluorescence from (**A**) was quantified. For each embryo, the fluorescence was normalized to its intensity at 0 min, and the fraction remaining is plotted through the time course. Significance was calculated using the Mann-Whitney test. (**C**) GNU is required for the destruction of ME31B. dsRNA targeting *GNU* or mCherry (as a control) mRNA was expressed during oogenesis; female flies also contained the ME31B-GFP allele. Laid embryos were visualized by fluorescence microscopy at indicated time points, otherwise as in A. (**D**) PLU is required for the destruction of ME31B. As in C, except with dsRNA targeting *PLU* mRNA. (**E**) Fertilization is not required for the destruction of ME31B. As in A, except with fertilized and unfertilized embryos.

## ME31B is ubiquitinated

We hypothesized that ME31B degradation involved ubiquitination. To test this model, we immuno-precipitated ME31B-GFP from 1 to 2 hr embryo lysates under stringent conditions that disrupted protein-protein interactions, such as that with eIF4E. An ubiquitin smear was detected in immuno-precipitants by western blotting at a size consistent with polyubiquitinated ME31B-GFP (*Figure 2A*). We also detected ubiquitin by mass spectrometry of ME31B-GFP pull-downs (see below). This

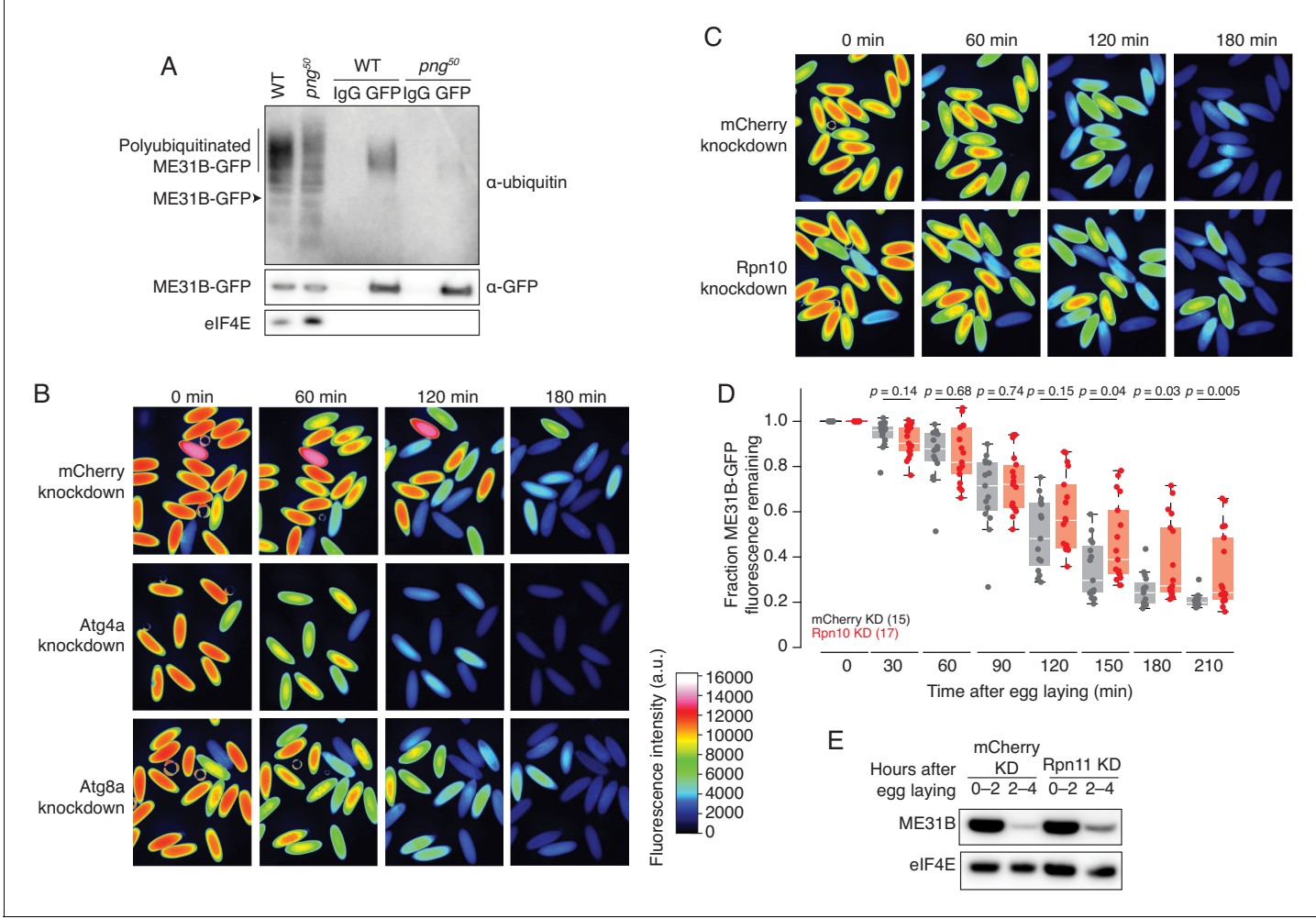

**Figure 2.** ME31B is degraded by the ubiquitin-proteasome pathway. (**A**) Ubiquitination of ME31B-GFP requires the PNG kinase. ME31B-GFP was immunoprecipitated under stringent conditions from 1–2 hr lysates from wild-type or *png*[50] embryos, and then analyzed by western blotting by probing with α-GFP, α-eIF4E or α-ubiquitin. (**B**) ME31B-GFP is not degraded by autophagy. Embryos from female flies expressing the indicated dsRNAs and ME31B-GFP were imaged at various time points by fluorescence microscopy. Fluorescent images are false-colored so that more intense fluorescence is indicated by hotter colors; fluorescence scale is shown. (**C**) Inhibiting the proteasome partially stabilizes ME31B-GFP. As in B, except for dsRNAs targeting mCherry or *Rpn10*, a proteasome component. (**D**) ME31B-GFP is stabilized in Rpn10-depleted embryos. ME31B-GFP fluorescence from (**C**) and intervening time points was quantified. For each embryo, the fluorescence was normalized to its intensity at 0 min, and the fraction remaining is plotted through the time course. Significance was calculated using the Mann-Whitney test. (**E**) Depleting the proteasome partially stabilizes ME31B-GFP. Staged embryos from the indicated times were harvested with mCherry or Rpn11 knocked down. Western blotting was performed on the lysates, probing for GFP and eIF4E (as a loading control). Related to Source *Figure 2—source data 1*.

The online version of this article includes the following source data for figure 2:

**Source data 1.** Summary of RNAi screen.

ubiquitination was not detectable in *png*[50] mutant embryos and thus depended upon the PNG kinase activity (*Figure 2A*).

We next asked whether ME31B-GFP is degraded by autophagy or by the proteasome. To do so, we depleted components of either system using the GAL4-UAS system described above. Knockdown of five autophagy components, such as Atg4a or Atg8a, gave viable embryos. However, ME31B-GFP was not stabilized in any of these knockdown embryos; indeed, in some cases, it appeared to be degraded more quickly than in the control embryos (*Figure 2B*, *Figure 2—source data 1*).

Analysis of embryos depleted of core barrel proteasome proteins proved more challenging because knockdown of most components, such as Prosα5 and Prosα7, resulted in females that did not lay eggs (*Figure 2—source data 1*). We were able to obtain embryos from *Rpn10* and *Rpn11* knockdowns, two components of the regulatory particle of the proteasome, perhaps because these embryos only had partial inhibition of proteasome function or there is functional redundancy. Importantly, we observed partial stabilization of ME31B-GFP in both knockdown embryos (*Figure 2C–E*). The role of the proteasome in degrading ME31B is consistent with results from a complementary study where injection of MG132 into embryos stabilized endogenous ME31B, TRAL, and Cup (*Cao et al., 2019*). Thus, taken together, these data suggest that the ubiquitin-proteasome system degrades ME31B.

## The E2 conjugating enzyme UBC-E2H/Marie Kondo is required for the degradation of ME31B, TRAL, and Cup

We thus set out to identify E2 conjugating enzymes and E3 ligases responsible for the degradation of ME31B by carrying out a medium-scale RNAi screen. As before, we monitored ME31B-GFP decay by GFP fluorescence, taking images every 30 min after egg laying. We focused on those proteins that: (1) had evidence of expression, based on RNA-seq or mass spectrometry data, and (2) had available RNAi lines. We screened 137 RNAi lines, targeting E3 ligases as well as related factors (*Figure 2—source data 1*). Note that RNAi lines knocking down many cullins and proteasomal components did not lay eggs, presumably because of critical functions during oogenesis. Because we did not measure mRNA or protein levels in the screen or in subsequent experiments (due to COVID-19 restrictions), we do not know the efficiency of RNAi knock-down. Although our initial E3 screen failed to reveal any strong candidates, knockdown of UBC-E2H, an E2 ligase conserved from yeast to humans (*Kaiser et al., 1995*; *Kaiser et al., 1994*; *Lampert et al., 2018*), blocked degradation of ME31B-GFP and nearly phenocopied the dynamics seen in $png^{50}$ mutants (*Figure 3A*).

To test this result, we raised an antibody against UBC-E2H and confirmed that the protein was depleted in the knockdown embryos (*Figure 3B*). We next isolated lysates from staged embryos and performed western blotting, probing for ME31B. Because the maternal line contains both the wild-type and trap ME31B alleles, this experiment revealed that both wild-type and GFP fusion proteins were stabilized when UBC-E2H was depleted (*Figure 3C*), albeit more so for the fusion protein than wild-type one. Importantly, as determined by western blotting, endogenous, untagged Cup and TRAL were also stabilized in the UBC-E2H knockdown embryos (*Figure 3C*). Due to its role in removing proteins given in the maternal dowry, we renamed UBC-E2H as 'Marie Kondo' (shortened to 'Kdo'). Finally, through immunoprecipitation experiments, we determined that ubiquitination of ME31B-GFP was undetectable in *Kdo* knockdown embryos (*Figure 3D*). Thus, we conclude that Kdo is required for the destruction of ME31B, TRAL, and Cup during the MZT.

## Degradation of ME31B, TRAL, and Cup requires the CTLH E3 ligase

Kdo is conserved from yeast to humans and is known to work through the CTLH E3 ligase, a multi-component complex (*Lampert et al., 2018*; *Santt et al., 2008*). (Note that the *S. cerevisiae* complex is called the Gid complex.) Using BLAST for the human CTLH components, we were easily able to identify putative *D. melanogaster* homologs: RanBPM (homologous to *Hs* RanBP9), Muskelin, CG3295 (homologous to *Hs* RMND5A/GID2), CG7611 (homologous to *Hs* WDR26), CG6617 (homologous to *Hs* TWA1/GID8), and CG31357 (homologous to *Hs* MAEA) (*Figure 4A*). We were unable to find putative homologs for *Hs* GID5/ARMC8 or *Hs* GID4 (see below). Notably, none of these genes were annotated as putative E3 components in FlyBase, and thus none were included in our original screen.

To ask if the CTLH complex might be involved in the degradation of ME31B, we immunoprecipitated ME31B-GFP and Cup-GFP from pre-MZT embryos (in conditions that maintain complex formation) and determined the proteins bound by mass spectrometry (*Figure 4B and C*; *Figure 4—source datas 1* and *2*). Consistent with our previous work (*Wang et al., 2017*), both immunoprecipitations readily pulled-down other members of the Cup–TRAL–ME31B complex. We were able to identify Muskelin, RanBPM, and CG6617 in all four samples. We also detected CG3295 in both ME31B-GFP pull-downs and one Cup-GFP pull-down, and CG31357 in both ME31B-GFP pull-downs. Similar results were seen in previous studies of ME31B complexes in embryonic lysates (*Götze et al.,*

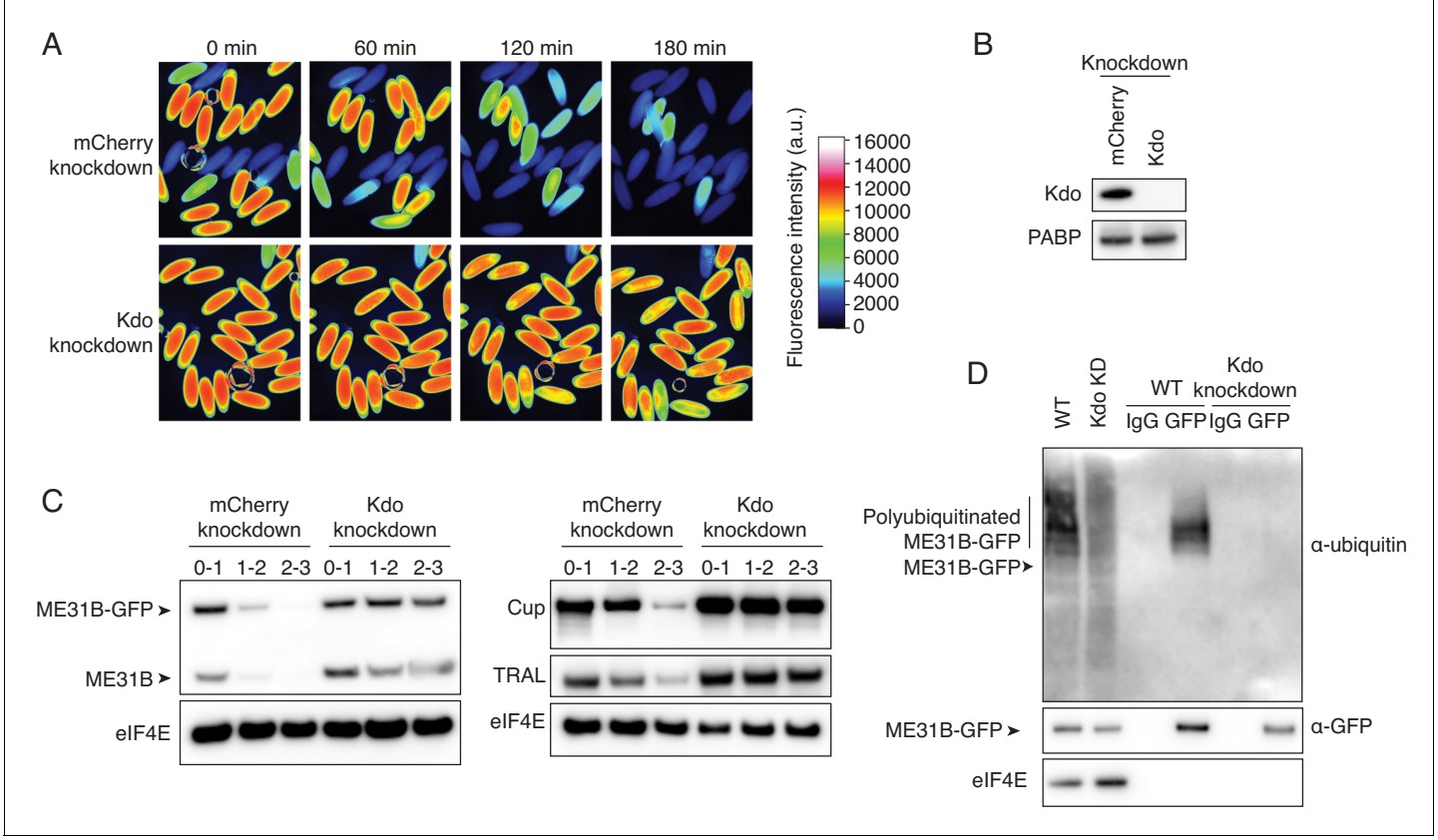

**Figure 3.** The Marie Kondo E2 conjugating enzyme mediates the degradation of ME31B, TRAL, and Cup. (**A**) Marie Kondo (Kdo/UBC-E2H) is required for the degradation of ME31B-GFP. Embryos laid from female flies expressing the indicated dsRNAs and ME31B-GFP were imaged at various time points by fluorescence microscopy. Fluorescent images are false-colored so that more intense fluorescence is indicated by hotter colors; fluorescence scale is shown. (**B**) Kdo is depleted in knockdown lines. An antibody was raised against Kdo, and 1–2 hr embryo lysates with the indicated dsRNA were probed for Kdo and PABP (as a loading control). (**C**) Kdo is required for destruction of ME31B, TRAL, and Cup. Staged embryos with mCherry or Kdo knocked down were harvested at the indicated times. Western blotting was performed on the lysates, probing for ME31B, Cup, TRAL, and eIF4E (as a loading control). Because the maternal flies contained wild-type and ME31B-GFP alleles, both proteins are visible in the α-ME31B blot. (**D**) Kdo is required for the ubiquitination of ME31B. ME31B-GFP was immunoprecipitated from 1 to 2 hr mCherry or Kdo knock-down embryo lysates under stringent conditions and then analyzed by western blotting by probing with α-GFP, α-eIF4E, or α-ubiquitin. IgG was used as an immunoprecipitation control.

2017) and in a complimentary study (*Cao et al., 2019*). We were unable to detect CG7611 in any of our samples.

We next asked whether destruction of ME31B requires the CTLH E3 ligase. As before, we knocked down various components using available RNAi lines (CG3295, CG7611, Muskelin, and RanBPM), and monitored levels of ME31B-GFP by fluorescence microscopy (*Figure 4D*). In contrast to what we had observed with other E3 ligases (*Figure 2—source data 1*), depletion of CG3295, Muskelin, or RanBPM almost completely stabilized ME31B-GFP. RNAi directed against CG7611, the one component that we failed to detect by mass spectrometry, did not affect the destruction of ME31B, although we cannot exclude the possibility that CG7611 protein levels were not sufficiently affected. Consistent with these results, when we used western blotting to look at levels of ME31B, Cup and TRAL, we found that all were stabilized when the CTLH complex was depleted (*Figure 4E*).

One trivial explanation for these results is that depletion of the CTLH complex inadvertently reduced levels of Kdo, which is required for the destruction of ME31B (*Figure 3*). However, as determined by western blotting, Kdo levels were unaffected in these knockdown embryos (*Figure 4—figure supplement 1A*). Because antibodies were only available for RanBPM (*Dansereau and Lasko, 2008*), we were unable to generally determine how depletion of individual components affected

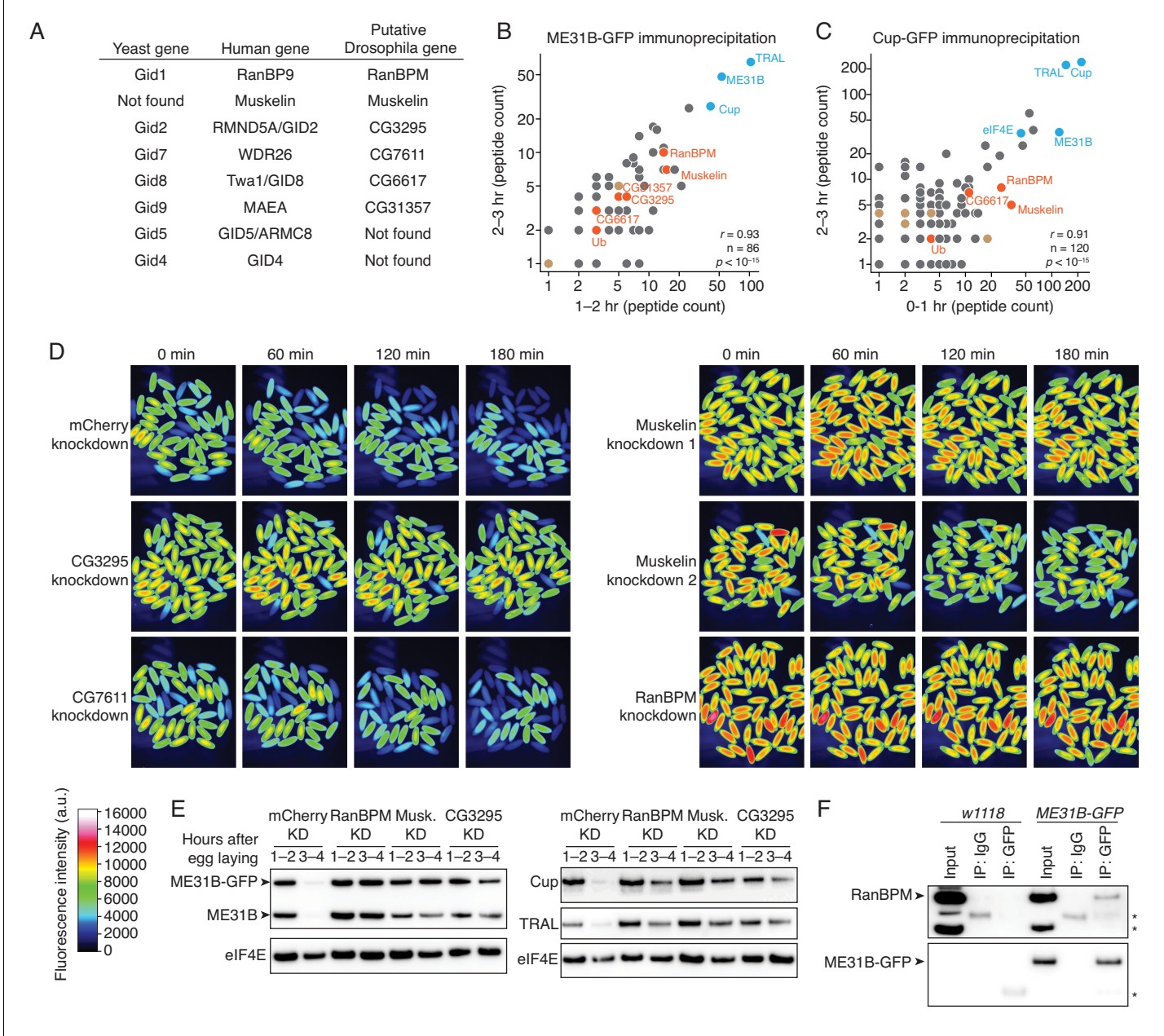

**Figure 4.** The CTLH E3 ligase mediates the degradation of ME31B, TRAL, and Cup. (**A**) Putative components of the *Drosophila* CTLH complex. *Drosophila* orthologs of the human components of the CTLH complex were determined by BLAST. Some components, such as GID4, appear to not have an identifiable ortholog. (**B**) ME31B interacts with the CTLH complex. Proteins interacting with ME31B-GFP at 1–2 and 2–3 hr after egg laying were determined by immunoprecipitation with α-GFP and mass spectrometry. Plotted are number of peptides detected (after subtracting for the signal in the control IgG immunoprecipitation). Known interactors with ME31B, such as Cup and TRAL, are shown in blue; CTLH components and ubiquitin are shown in orange; other hits from the screen are shown in brown. (**C**) Cup interacts with the CTLH complex. As in B, except for Cup-GFP. (**D**) The CTLH complex is required for destruction of ME31B-GFP. Embryos laid from the indicated maternal RNAi lines were visualized for 3 hr after egg-laying for ME31B-GFP fluorescence. Fluorescent images are false-colored so that more intense fluorescence is indicated by hotter colors; fluorescence scale is shown. (**E**) The CTLH complex is required for destruction of ME31B, TRAL, and Cup. Staged embryos from the indicated times were harvested with mCherry, RanBPM, Muskelin (Musk.), or CG3295 knocked down. Western blotting was performed on the lysates, probing for ME31B, Cup, TRAL, and eIF4E (as a loading control). Because the maternal flies contained wild-type and ME31B-GFP alleles, both proteins are visible in the α-ME31B blot. (**F**) ME31B-GFP interacts with RanBPM, a component of the CTLH complex. ME31B-GFP complexes were immunoprecipitated from 1 to 2 hr embryo lysates using α-GFP or IgG (as a control). Membranes were probed with α-GFP or α-RanBPM. *, non-specific band.

The online version of this article includes the following source data and figure supplement(s) for figure 4:

**Source data 1.** ME31B immunoprecipitation/mass spectrometry results.

*Figure 4 continued on next page*

*Figure 4 continued*

**Source data 2.** Cup immunoprecipitation/mass spectrometry results.
**Figure supplement 1.** Depletion of CTLH components does not impact Kdo levels.

levels of the other components. Nonetheless, in analyzing RanBPM levels, we found that the RanBPM RNAi line was depleted for the protein (*Figure 4—figure supplement 1B*).

We next asked whether we could detect an interaction between ME31B and the CTLH complex using immunoprecipitation followed by western blotting. Consistent with our mass spectrometry analysis (*Figure 4B*), we detected RanBPM in ME31B-GFP immunoprecipitations (*Figure 4F*). Because we lack antibodies for other components, we were unable to probe interactions between ME31B and Muskelin, CG3295, and CG6617 by western blotting. As expected, when we performed control experiments in wild-type embryos, RanBPM was not immunoprecipitated (*Figure 4F*). Consistent with this result, we were able to immunoprecipitate RanBPM using antibodies recognizing endogenous Cup protein (see below).

Taken together, we conclude that the CTLH E3 ligase is required for the destruction of ME31B, Cup, and TRAL, and, in the early *Drosophila* embryo, is at least composed of RanBPM, Muskelin, CG6617, CG3295, and CG31357, although the role of CG7611 remains unknown. Because of their roles in clearing proteins, we also now refer to CG6617 as Houki (Hou, Japanese for 'broom'), CG3295 as Souji (Sou, Japanese for 'cleaning'), and CG31357 as Katazuke (Kaz, Japanese for 'tidying up').

## The *D. melanogaster* CTLH E3 ligase differs from the *S. cerevisiae* Gid complex

We next wanted to understand the organization of the *Drosophila* CTLH complex. Serendipitously, a cryoEM structure of the yeast Gid complex was recently published (*Qiao et al., 2020*). The Gid complex is composed of three sections: a catalytic module made by Gid2 and Gid9; a scaffold of Gid8, Gid1, and Gid5; and a substrate adaptor module formed by Gid4 (*Figure 5A*). By analogy, we were able to assign roles to the known *Drosophila* components: Kaz and Sou likely form the catalytic module, while RanBPM and Hou are part of the scaffold domain (*Figure 5A*). Consistent with such organization, when Sou was depleted, the interaction between RanBPM and ME31B-GFP was unaffected (*Figure 5B*). The interaction between RanBPM and ME31B-GFP did, however, depend upon Muskelin (*Figure 5B*), but with current structures, it is unclear how Muskelin interacts with the other components of the *Drosophila* CTLH complex.

In considering the organization of the *Drosophila* CTLH complex, we were surprised by the lack of any putative Gid5 or Gid4, which are important for the scaffold and substrate recognition, respectively. In yeast, Gid4 is exchangeable with other substrate adaptors, such as Gid10. Because the substrate adaptor module interacts with the core of the Gid complex predominantly through Gid5, we focused on identifying a *Drosophila* Gid5 ortholog. However, initial BLAST searches with both *S. cerevisiae* Gid5 and *H. sapiens* ARMC8 failed to identify a putative ortholog, and so we turned Phyre2 to conduct a 'BackPhyre' structure-homology-based search for a *Drosophila* Gid5 with the search model provided by the recent structure of the *S. cerevisiae* Gid complex (*Kelley et al., 2015*; *Qiao et al., 2020*). Although *Drosophila* proteins were identified that contained the armadillo domains (which is the major fold in Gid5), none of these were convincing hits to Gid5 overall, predicted to interact with Gid8, or identified in our mass spectrometry data (*Figure 4—source datas 1 and 2*; *Figure 5—source data 1*). In contrast, the same search performed against the human genome easily identified ARMC8, which was predicted to interact with Gid8 (*Figure 5—source data 1*).

Prompted by our continued inability to identify a *Drosophila* Gid5 ortholog, we threaded RanBPM and Hou into the *S. cerevisiae* structure so that we could examine the predicted Gid5 interface. These proteins broadly shared predicted structures with their yeast counterparts (*Figure 5C*). Despite overall predicted similarities with the yeast structure, we found two differences at the predicted interface between RanBPM–Hou and Gid5–Gid4 (*Figure 5D,E*). First, in *S. cerevisiae*, Gid4 makes the majority of its contacts only with Gid5. Nonetheless, one loop in Gid1 extends out to interact with Gid4 (*Qiao et al., 2020*). In contrast, no such loop is predicted in *Drosophila*

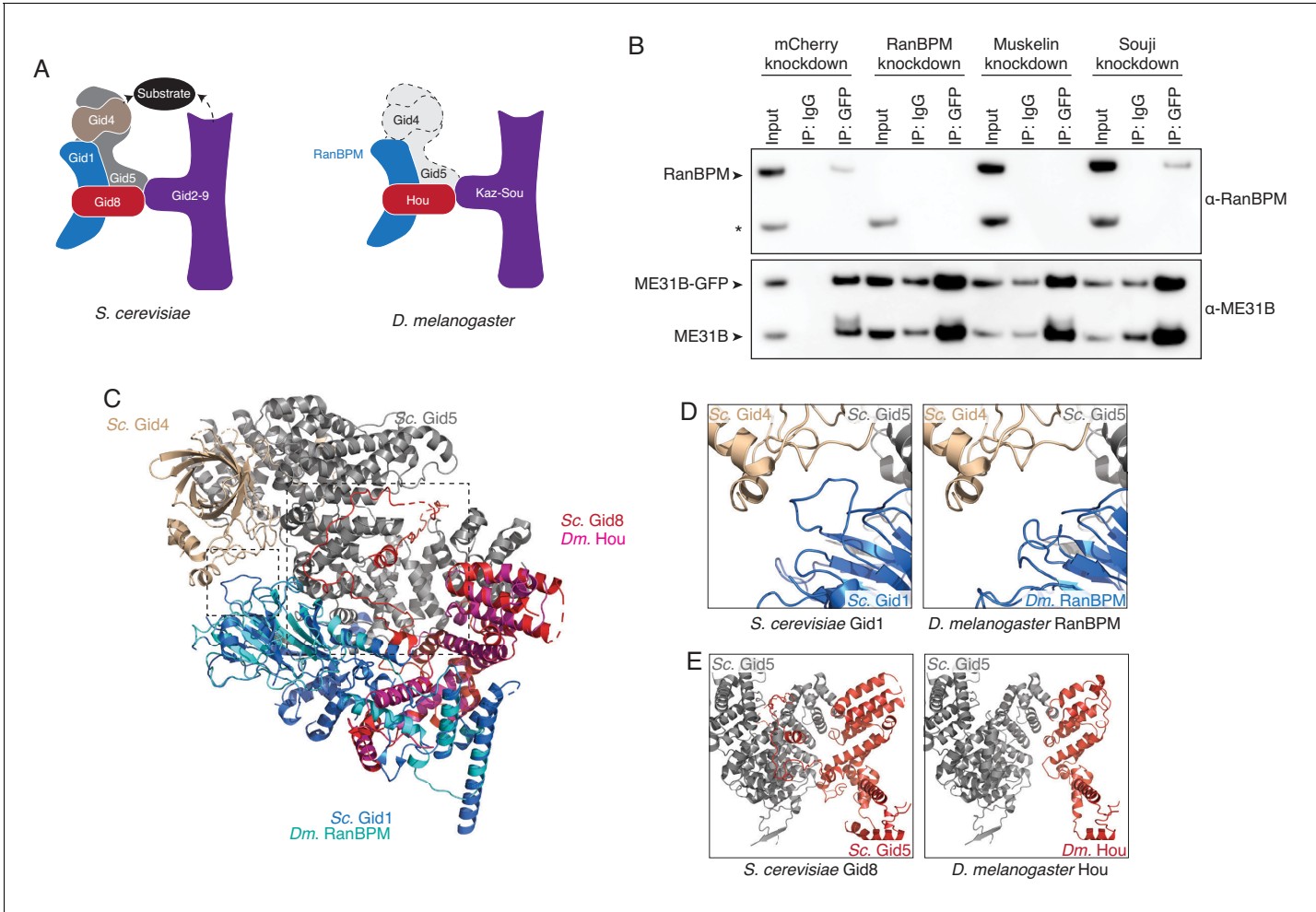

**Figure 5.** The *Drosophila* CTLH complex has a different organization than the *S. cerevisiae* Gid complex. (**A**) Schematic of the *S. cerevisiae* Gid complex based on recent cryo-EM structures (**Qiao et al., 2020**), and the *Drosophila* CTLH complex, based on orthology of the subunits. (**B**) The interaction between ME31B-GFP and RanBPM depends on Muskelin, but not Souji. ME31B-GFP complexes were immunoprecipitated from 1 to 2 hr embryo lysates using α-GFP or IgG (as a control). Immunoprecipitations were performed in 1–2 hr embryo lysates with mCherry, RanBPM, Muskelin, or Souji knocked-down. Membranes were probed with α-GFP or α-RanBPM. * indicates a non-specific band. (**C**) RanBPM and Hou share general structural similarities with their yeast orthologs. RanBPM (teal) and Hou (magenta) were threaded through structures of their orthologs, Gid1 (blue) and Gid8 (red), respectively. (**D**) RanBPM lacks the Gid4-interacting loop. Shown is the region of the *S. cerevisiae* Gid complex (left) and *D. melanogaster* CTLH complex (right) where Gid1 interacts with Gid4. The corresponding loop is missing in RanBPM. (**E**) Houki lacks the Gid5-interacting tail. Shown is the region of the *S. cerevisiae* Gid complex (left) and *D. melanogaster* CTLH complex (right) where Gid8 interacts with Gid5. The corresponding tail is missing in Houki. For simplicity, only Gid5 and Gid8 are shown.

The online version of this article includes the following source data and figure supplement(s) for figure 5:

**Source data 1.** Results of Gid5 Phyre2 search against the human and *Drosophila* genome.

**Figure supplement 1.** RanBPM and Houki lack key residues for CTLH complex formation.

RanBPM, although the flanking β sheet strands appear to exist (*Figure 5D*, *Figure 5—figure supplement 1A*). Second, in *S. cerevisiae*, the scaffold module is composed of Gid1, Gid5, and Gid8. Here, most of the Gid8 C-terminus wraps around Gid5 and makes nearly all of the interactions with Gid5. This domain appears to be also absent from *Drosophila* Hou, despite structural similarity in the rest of the protein (*Figure 5E*; *Figure 5—figure supplement 1B*). The consequence of *Drosophila* Hou lacking this domain is that there is little predicted interaction between Hou and Gid5. Taken together, these analyses suggest that substrate recognition is likely different for the *Drosophila* CTLH complex than in other organisms and leaves open the question of how ME31B, Cup, and TRAL bind the E3 ligase.

## Association between the CTLH E3 ligase and ME31B/Cup does not require PNG kinase activity

Having identified the E3 ligase and E2 conjugating enzyme, we next turned to understanding how the destruction of ME31B was triggered by PNG activity. Because recent work has demonstrated that the PNG kinase phosphorylates ME31B, TRAL, and Cup (*Hara et al., 2018*), we explored the idea that this phosphorylation might stimulate an association between ME31B and the E3 ligase.

Consistent with a role for the E3 ligase binding a target protein, interaction between RanBPM and ME31B-GFP was unaffected by Kdo depletion (*Figure 6A*). However, the interaction between RanBPM and ME31B-GFP remained robust in *png*$^{50}$ embryos (*Figure 6B*). Similarly, when we immunoprecipitated endogenous Cup, we were able to detect an interaction with RanBPM in both wild-type and *png*$^{50}$ embryos (*Figure 6C*). Thus, we conclude that PNG activity is not required for the CTLH complex to recognize and interact with ME31B and Cup.

## Egg activation mediates the translational upregulation of *Kdo* via the PNG kinase

Given that PNG phosphorylation of ME31B could not explain how egg activation stimulated its association with the CTLH complex, we searched for alternative explanations, focusing on recent observations that PNG also mediates the translational upregulation of thousands of mRNAs at the oocyte-to-embryo transition (*Eichhorn et al., 2016*). We analyzed published ribosome profiling datasets (*Eichhorn et al., 2016*) for evidence of translational upregulation of CTLH components and *Kdo* mRNAs upon the oocyte-to-embryo transition. Known CTLH component mRNAs were either not affected or downregulated during egg activation (*Figure 7—figure supplement 1*), although we cannot exclude the hypothesis that unidentified components may be regulated by PNG.

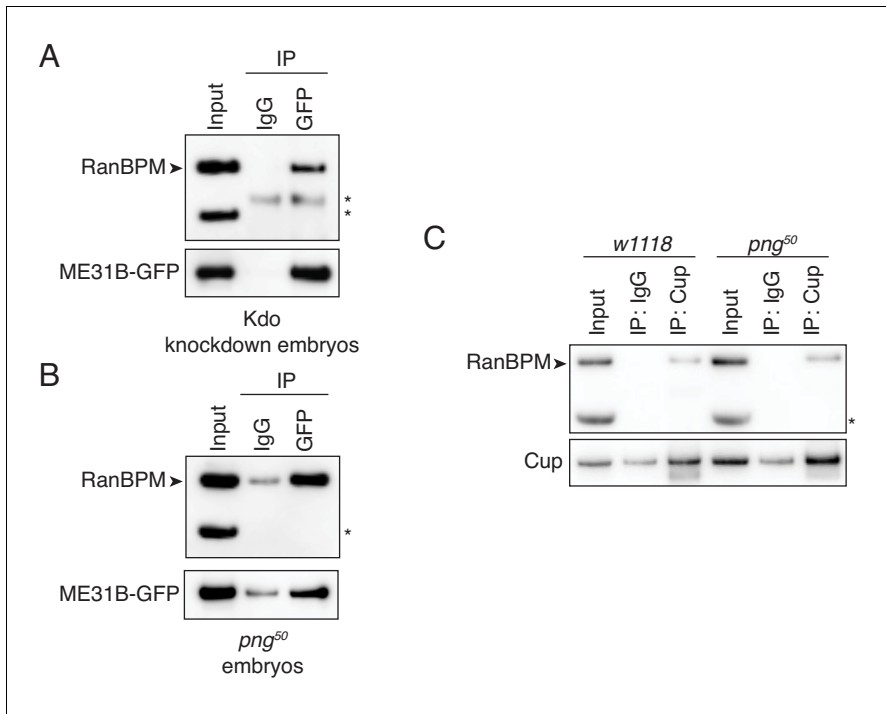

**Figure 6.** The interaction between ME31B and the CTLH complex does not depend on PNG. (**A**) The interaction between ME31B-GFP and RanBPM does not depend on Kdo. ME31B-GFP complexes were immunoprecipitated from 1–2 hr embryo lysates using α-GFP or IgG (as a control). Immunoprecipitations were performed in 1–2 hr embryo lysates with Kdo knocked down. *, non-specific band. (**B**) The interaction between ME31B-GFP and RanBPM does not depend on PNG. As in B, except in 1–2 hr *png*$^{50}$ embryo lysates. *, non-specific band. (**C**) The interaction between Cup and RanBPM does not depend on PNG. Complexes were immunoprecipitated from 1–2 hr lysates from wild-type (*w1118*) or *png*$^{50}$ embryos using α-Cup or IgG. Membranes were probed with α-Cup or α-RanBPM. *, non-specific band.

In contrast, the most striking change occurred for translation of *Kdo* mRNA: although translation of *Kdo* mRNA was repressed through oogenesis, its translation increased 25-fold during the oocyte-to-embryo transition (*Figure 7A*), placing it in the top 10% of genes upregulated at this developmental transition. However, *Kdo* was not translationally upregulated in *png50*-activated embryos (*Figure 7B*), and its translation differed by more than 200-fold between wild-type and mutant-activated eggs. In fact, *Kdo* was the seventh-most affected transcript (*Figure 7C*), showing a similar

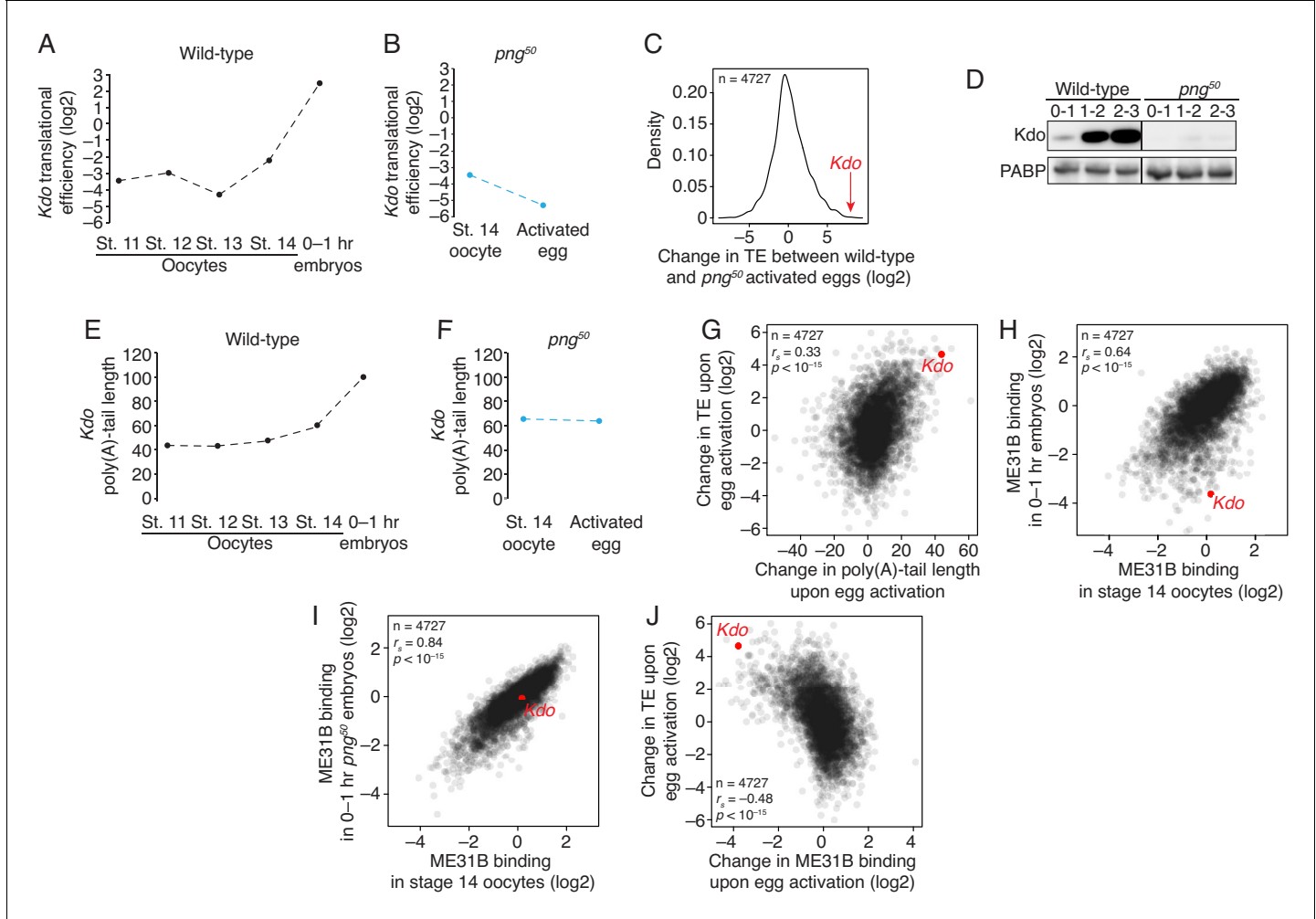

**Figure 7.** The PNG kinase mediates translational upregulation of *Kdo* at egg activation. (A) Translation of *Kdo* increases at egg activation. Translational efficiency of *Kdo* mRNA was measured in published ribosome profiling datasets (*Eichhorn et al., 2016*). Shown is the translational efficiency in stage 11, 12, 13, and 14 oocytes, and 0–1 hr embryos. (B) Translational upregulation of *Kdo* depends on PNG. As in A, except for *png50* stage 14 oocytes and activated eggs. (C) Translation of *Kdo* is highly dependent on PNG. Shown is a density plot of the difference in translational efficiency between wild-type and *png50* embryos. The difference in translational efficiency for *Kdo* mRNA is shown by an arrow. (D) Production of Kdo during embryogenesis depends on PNG. Staged wild-type or *png50* embryos were harvested at the indicated times after egg laying. Western blotting was performed on the lysates, probing for Kdo and PABP (as a loading control). (E) *Kdo* mRNA poly(A)-tail length increases at egg activation. As in A, except plotting poly(A)-tail lengths from Eichhorn, et al. (F) Lengthening of *Kdo* mRNA poly(A)-tail length depends on PNG. As in B, except plotting poly(A)-tail length. (G) Changes in poly(A)-tail length during egg activation correlate with changes in translation efficiency. Shown is scatter plot comparing, for each gene, the change in poly(A)-tail length between stage 14 oocytes and 0–1 hr embryos and the change in its translational efficiency. *Kdo* is highlighted in red. (H) ME31B changes upon egg activation. Shown is a scatter plot comparing ME31B binding to mRNAs in stage 14 oocytes and 0–1 hr embryos. *Kdo* is highlighted in red. (I) Changes in ME31B binding upon egg activation depend on PNG. As in H, except comparing ME31B binding in stage 14 oocytes and 0–1 hr *png50* embryos. (J) Changes in ME31B at egg activation correlate with changes in translational efficiency. Shown is a scatter plot comparing changes in ME31B binding between stage 14 oocytes and 0–1 hr embryos and changes in translational efficiency. *Kdo* is highlighted in red.

The online version of this article includes the following figure supplement(s) for figure 7:

**Figure supplement 1.** Translation of CTLH components does not increase during the oocyte-to-embryo transition.

dependence on PNG as *Smaug*, which encodes a well-known and important downstream target of PNG activity.

Consistent with this analysis, when we probed Kdo protein levels, we were able to detect a dramatic increase in protein levels over the first 3 hr of embryogenesis in wild-type embryos, but we were unable to detect expression at any time point in *png*[50] mutant embryos (*Figure 7D*). Pointing to a role of post-transcriptional gene regulation, the poly(A) tail length on *Kdo* mRNA also doubled upon egg activation in a manner dependent on PNG. This result suggests that the translational increase is partially mediated by an increase in poly(A)-tail length (*Figure 7E,F*), which directly impacts translational changes during egg activation (*Eichhorn et al., 2016*).

We noted, however, that although a change in poly(A)-tail length partially explained the changes in translational efficiency (*Figure 7G*), it did not fully account for all of the translational changes at egg activation. Because we had previously found that ME31B is broadly associated with translational repression in the early embryo (*Wang et al., 2017*), we wondered if ME31B binding might also change during egg activation and if such changes could impact translation. To test this possibility, we immunoprecipitated ME31B-GFP complexes from stage 14 oocytes and sequenced bound transcripts, as we have done previously (*Wang et al., 2017*). We normalized bound RNA to total abundance to generate an overall occupancy. Although ME31B-GFP binding in stage 14 oocytes was highly correlated with that measured previously in 0–1 hr wild-type embryos (Spearman r [$r_s$]=0.66, p<$10^{-15}$), its binding was even more similar in 0–1 hr *png*[50] embryos ($r_s$ = 0.84, p<$10^{-15}$; Fisher's r-to-z transformation: p<$10^{-15}$; *Figure 7H,I*). These data indicate that although ME31B binds broadly similar transcripts in the oocyte and embryo, its binding does change at egg activation in a manner dependent on the PNG kinase. Moreover, we found that the change in ME31B-GFP binding at egg activation was strongly correlated with the change in translational efficiency ($r_s$ = –0.48, p<$10^{-15}$) such that those mRNAs with diminished ME31B-GFP binding were translationally activated and those with increased binding were translationally repressed (*Figure 7J*). Importantly, this relationship held even after we controlled for changes in poly(A)-tail length ($r_s$ = –0.40, p<$10^{-15}$). Thus, these analyses indicate that PNG activity at egg activation triggers two independent, albeit related, mechanisms for altering translation: changing poly(A)-tail length and altering ME31B binding.

Importantly, as with poly(A)-tail length, ME31B-GFP binding to *Kdo* mRNA also changed upon egg activation and was substantially reduced in a PNG-dependent manner (*Figure 7H,I*). Taken together, these data suggest that, during oogenesis, ME31B, presumably via Cup and TRAL, acts to repress translation of *Kdo* mRNA and thus suppress production of its own E2. Upon egg activation, PNG activity not only leads to extension of the *Kdo* mRNA poly(A) tail, but also stimulates the dissociation of ME31B from the transcript. Together, these two activities likely promote translation of *Kdo*, setting the stage for clearance of deposited ME31B, TRAL, and Cup.

## Discussion

ME31B, Cup, and TRAL are RNA-binding proteins that are degraded during the MZT. Despite occurring several hours after egg laying, degradation of these proteins is triggered by egg activation through the activity of the PNG kinase and appears to be mediated by the ubiquitin-proteasome system (*Figures 1* and *2*). Through a medium-scale RNAi screen, we identified that the E2 conjugating enzyme Kdo is required for the clearance of ME31B, TRAL, and Cup (*Figure 3*). Kdo is conserved from yeast to humans and, as in those systems (*Kaiser et al., 1994*; *Lampert et al., 2018*), appears to work with the CTLH complex, which acts as the E3 ligase. Components of the CTLH complex physically interact with ME31B and Cup, and the CTLH complex is also required for the degradation of ME31B, TRAL, and Cup during early embryogenesis (*Figure 4*). Structure-based homology suggests that, despite its conservation from yeast to humans, the *Drosophila* CTLH complex has an unusual architecture, and it remains unclear how it recognizes its substrate (*Figure 5*). The association of CTLH with ME31B occurs in the absence of PNG activity, suggesting that, although ME31B (as well as TRAL and Cup) are phosphorylated by the kinase, phosphorylation may not be required for their destruction (*Figure 6*). Instead, translation of *Kdo* appears to be suppressed during oogenesis by its short poly(A) tail length and binding of ME31B. Its translation is dramatically upregulated at the oocyte-to-embryo transition, in a process that depends on PNG activity (*Figure 7*). Together, these data suggest a model that egg activation via the PNG kinase leads to translational activation and production of Kdo, which then allows the CTLH complex to ubiquitinate ME31B, TRAL, and Cup

and ultimately leads to their destruction (*Figure 7K*). Interestingly, based on RNA-seq data from Fly-Base (*FlyBase Consortium et al., 2019*), Muskelin shows exquisite tissue-specificity and is only strongly expressed in the ovaries. This observation, together with the translational control of *Kdo*, may partly explain how ME31B, a ubiquitous protein, is specifically destabilized in the early embryo.

Although the CTLH complex is conserved, it has not yet been studied in *Drosophila*. Our data point to this complex being composed of multiple components (Muskelin, RanBPM, Houki, Souji, and Katazuke), as in other organisms. However, due to a lack of available reagents, we do not know about the stoichiometry of these components, and it remains possible that there are additional, *Drosophila*-specific components. Nonetheless, so far, the CTLH complex in *Drosophila* appears different than the human and yeast complexes. Although Gid7 and WDR26 are important in the yeast and human versions, respectively, and we identified a *Drosophila* ortholog (CG7611), we found no evidence of its association with ME31B or requirement for ME31B degradation; the role of CG7611 in the *Drosophila* CTLH complex warrants further investigation. We were also unable to identify orthologs of Gid4 and Gid5, which are critical for substrate recognition in *S. cerevisiae* (*Lampert et al., 2018*; *Maitland et al., 2019*; *Santt et al., 2008*). Intriguingly, the residues and domains important for the Gid1–Gid4 and Gid8–Gid5 interactions in budding yeast appear absent to be in RanBPM and Hou, raising the fundamental question of how the *Drosophila* CTLH complex recognizes and positions its substrate proteins. Answering this question will require future investigation and may shed light on other proteins targeted by the *Drosophila* CTLH complex and the extent to which ME31B, a ubiquitously expressed protein, is targeted outside of the MZT.

One unexpected result is the role of PNG in mediating the destruction of ME31B, TRAL, and Cup. PNG phosphorylates all three proteins (*Hara et al., 2018*), and so our initial hypothesis was that this modification also stimulated their destruction. However, contrary to our expectations, ME31B and Cup interacted with the CTLH complex even in $png^{50}$ embryos, demonstrating that phosphorylation by PNG was not required for binding of ME31B and Cup by the E3 ligase. An unresolved question, then, is how PNG phosphorylation affects the activities of ME31B, TRAL, and Cup. Intriguing observations from the Orr-Weaver lab suggest that the modification can impact the ability of these proteins to repress gene expression (*Hara et al., 2018*). It is tempting to speculate that phosphorylation may then contribute to the MZT by modulating the activities of ME31B, TRAL, and Cup, rather than their stability.

The link between PNG and the destruction of ME31B, TRAL, and Cup instead appears to be mediated through the translational upregulation of *Kdo*. Although PNG may contribute through other, as-yet undiscovered, mechanisms as well (such as phosphorylating unknown CTLH adaptor proteins), this link is sufficient to explain the PNG requirement for ME31B degradation: in the absence of Kdo, ME31B is stable during the MZT, and in the absence of PNG, Kdo is not detectably expressed. An important question for the future will be to understand what elements in the *Kdo* mRNA are responsible for its translational repression during oogenesis. One hint may be that the 3'UTR of *Kdo* contains several putative Pumilio-binding sites, and translation of *Kdo* is upregulated in ovaries where Pumilio has been knocked down (*Flora et al., 2018*). Pumilio is also a target of PNG (*Hara et al., 2018*), and so a possible model is that translational repressors, such as Pumilio, are phosphorylated and inactivated at egg activation, leading to the production of Kdo.

PNG also mediates the translational upregulation of key MZT effectors: Zelda, the pioneer transcription factor, and Smaug, an RNA-binding protein that targets nearly two-thirds of the maternal transcriptome for degradation (*Chen et al., 2014*; *Eichhorn et al., 2016*; *Tadros et al., 2007*). Together with our results, a picture is emerging that egg activation stimulates the production of multiple key factors that are important for clearing the maternal RNA and protein dowry and for producing zygotic gene products.

Although the MZT has typically been considered from the perspective of RNA, a role for maternal protein clearance is becoming clearer. Over the past few years, the list of proteins degraded during the *Drosophila* MZT has grown and now includes GNU, Matrimony, Cort, Smaug, ME31B, TRAL, and Cup (*Benoit et al., 2009*; *Hara et al., 2017*; *Pesin and Orr-Weaver, 2007*; *Wang et al., 2017*; *Whitfield et al., 2013*). Unbiased mass spectrometry experiments also suggest that Wispy and Dhd are also robustly degraded (*Sysoev et al., 2016*). As this list of proteins in *Drosophila* and other developmental systems increases, a new question is emerging: how many maternally deposited proteins are degraded during the MZT? Understanding the mechanisms controlling protein degradation

during the MZT as well as the impact of removing the maternal protein dowry will be key issues to explore in the future.

## Materials and methods

### *Drosophila* fly stocks

Fly stocks were maintained in a 25°C incubator with 65% humidity.

### Microscopy

Male flies from the TRiP stocks were crossed with the ME31B-GFP driver line. Female flies from this cross were collected and crossed with *w1118* males. For egg collection, flies were transferred in the morning to egg-collection chambers on apple juice/agar plates. Flies were allowed to lay eggs for 1 hr. Eggs were collected into cell strainers, washed with 1X PBS, dechorionated with 25% bleach, and washed with 1X PBS. Dechorionated eggs were transferred onto a glass slide and covered with halo-carbon oil 700 (Sigma). Images were taken on a ZEISS SteREO Discovery.V8 microscope with X-Cite 120Q fluorescence illumination system (Exelitas Technologies). For the RNAi screen, phenotypes were scored qualitatively (*Figure 2—source data 1*). Quantitation of images was performed using ImageJ, and the data were subsequently processed in R using in-house scripts.

### Isolation of stage 14 oocytes

Stage 14 oocytes were isolated from a large-scale *Drosophila* culture established with $w^{1118}$ flies and were homogenized in lysis buffer B (with protease inhibitor cocktail (BioShop) and additional freshly added protease inhibitors [100 µM Leupeptin, 100 µM Chymostatin, 4 mM Benzamidine HCl, 3 µM Pepstatin; Sigma] and SUPERase-In RNase inhibitors). The homogenized lysates were clarified at 15,000 rpm, 4°C for 15 min, and the supernatant was stored at –80°C.

### Isolation of embryos

Embryos were collected at various time points post-egg laying, dechorionated with bleach, and washed with 0.1% Triton X-100. Embryos were then homogenized in lysis buffer B (150 mM KCl, 20 mM HEPES-KOH pH 7.4, 1 mM MgCl$_2$, 1 mM DTT, complete mini EDTA-free protease inhibitors), and were clarified at 15,000 rpm, 4°C for 15 min. The supernatant was stored at –80°C.

### Western blotting

The rabbit anti-Kdo antibody was generated by Pacific Immunology. For western blotting, it was used at 1:10,000. Rat anti-Cup (a gift of C. Smibert) was used at 1:5000. Mouse anti-ME31B antibody (a gift of K. Nakamura) was used at 1:5000. Rabbit anti-TRAL (a gift of K. Nakamura) was used at 1:5000. Rabbit anti-eIF4E (a gift of E. Izaurralde) was used at 1:10,000. Rabbit anti-PABP (a gift of E. Izaurralde) was used at 1:10,000. Rabbit anti-RanBPM (a gift of P. Lasko) was used at 1:10,000. Mouse anti-GFP (Roche) was used at 1:1000. Mouse anti-ubiquitin (ThermoFisher Scientific) was used at 1:5000.

### Immunoprecipitations

For immunoprecipitations to probe for interactions with Cup or ME31B-GFP, pre-made lysates (described above) were diluted to 1.0 mg/ml, and then incubated with anti-GFP (Roche) or rabbit IgG (Abcam) for 1 hr, rotating at 4°C. EZView protein G affinity beads (Sigma) were washed 3X with lysis buffer, and 25 µl of slurry was added to the lysate-antibody mixture and incubated for 1 hr, rotating at 4°C. Beads were washed three times with lysis buffer and transferred to a new tube. For western blot analysis, the beads were boiled in loading sample buffer and reducing agent, and immunoprecipitates were loaded onto an SDS-PAGE gel. For ME31B and RanBPM, 2.4% input and 17% IP were loaded.

For immunoprecipitations to test for ubiquitination, pre-made lysates (from above) were incubated at 4°C with anti-GFP or rabbit IgG for 1 hr, rotating. EZView protein G affinity beads were instead washed 2X with RIPA buffer, and then 25 µl of the slurry was added to the lysate-antibody mixture. The lysate-antibody mixture with the beads was diluted with RIPA buffer supplemented with 50 µM PR-619 and protease inhibitors, then incubated for another hour at 4°C, rotating. After

incubation, the beads were washed three times with supplemented RIPA buffer and transferred to a new tube. Beads were boiled in loading sample buffer and reducing agent, and immunoprecipitates were loaded on an SDS-PAGE gel. To probe for ubiquitin, 3% input and 20% IP were loaded; to probe for GFP, 1% input and 10% IP were loaded; to probe for eIF4E, 3% input and 10% IP were loaded.

For RNA immunoprecipitation, the beads were blocked overnight with salmon ssDNA and incubated with anti-GFP antibodies. Stage 14 oocyte lysates were diluted to 1.1 mg/ml, and then incubated with conjugated beads for 2–3 hr, rotating at 4°C. Beads were washed six times with the lysis buffer.

## Tandem mass spectrometry

Immunoprecipitates were sent to SPARC BioCentre (SickKids) for LC/MS/MS analysis as described previously (*Wang et al., 2017*). Briefly, immunoprecipitates were reduced with 10 mM DTT and then treated with 10 mM iodoacetemaide. Samples were digested in solution with trypsin. Mass spectrometry was performed on Q Exactive with dynamic exclusion. Peptides were searched with Sequest against the *Drosophila* uniprot database, and spectral counts were reported. Ubiquitin peptides were manually identified by mapping to ubiquitin-regions of fusion genes.

## RNA sequencing

RNA was extracted from immunoprecipitates and input lysate using TRI-reagent. To assess the enrichment of the RNA-immunoprecipitation, a fraction of the RNA was treated with DNase and used for RT-qPCR to ensure enrichment of *Act5C* transcripts. After verifying the quality of the RNA-immunoprecipitation, the RNA was subjected to Ribo-Zero Gold rRNA depletion according to the manufacturer's protocol. Libraries were then generated using Illumina's TruSeq stranded mRNA library preparation kit according to the manufacturer's protocol and sequenced at The Center for Applied Genetics (SickKids).

## Computational analyses

Libraries were pooled and sequenced on an Illumina HiSeq 2500 by The Centre for Applied Genomics at The Hospital for Sick Children. 50 base-pair single-end reads were demultiplexed and converted to FASTQ format using bcl2fastq2 v2.17 (Illumina). Library quality was inspected using FastQC v0.11.5 (http://www.bioinformatics.babraham.ac.uk/projects/fastqc/). Reads were trimmed for quality and clipped for Illumina adaptors using TrimmomaticSE version 0.36 (*Bolger et al., 2014*). Surviving reads were mapped by STAR 2.5.2a (*Dobin et al., 2013*) to the *D. melanogaster* genome obtained from UCSC on 7 August 2016. Genes were quantified using Cufflinks 2.2.1 (*Trapnell et al., 2010*). Downstream analyses were then performed with R version 3.1.2, using in-house scripts. Occupancies were calculated for each gene by dividing the IP FPKM by the input FPKM. When calculating occupancy, all the genes were filtered such that only genes with greater than 0.5 FPKM were included in the analysis. High-throughput sequencing data described in this paper are available from the GEO: GSE83616 (*Eichhorn et al., 2016*), GSE98106 (*Wang et al., 2017*), and, for the data prepared in this paper, GSE140436.

## Protein structure predictions and visualizations

RanBPM and Hou were visualized using standard Phyre2 parameters (*Kelley et al., 2015*). Outputs were visualized using Pymol and compared with the Gid complex (*Qiao et al., 2020*).

## Acknowledgements

We thank Dr. Julie Claycomb, Dr. Suja Jagannathan, Dr. Howard Lipshitz, Dr. Tania Reis, and members of the Rissland lab for helpful and thoughtful discussions. We thank Dr. Jeffrey Kieft for thought-provoking structure discussions. We also thank Dr. Howard Lipshitz and Dr. Elmar Wahle for sharing unpublished results. We thank Laura White for the original suggestion of "Marie Kondo" for the name of an E2.

Support was provided from the University of Colorado RNA Bioscience Initiative (OSR) and NIH grant R35GM128680 (OSR).

## Additional information

### Funding

| Funder | Grant reference number | Author |
|--------|------------------------|--------|
| National Institute of General Medical Sciences | R35GM128680 | Olivia S Rissland |
| University of Colorado | | Olivia S Rissland |

The funders had no role in study design, data collection and interpretation, or the decision to submit the work for publication.

### Author contributions

Michael Zavortink, Formal analysis, Investigation, Methodology, Writing - review and editing; Lauren N Rutt, Data curation, Formal analysis, Investigation, Methodology, Writing - original draft, Writing - review and editing; Svetlana Dzitoyeva, Formal analysis, Investigation, Methodology, Writing - original draft, Writing - review and editing; Jesslyn C Henriksen, Investigation, Writing - review and editing; Chloe Barrington, Danielle Y Bilodeau, Investigation, Methodology, Writing - original draft, Writing - review and editing; Miranda Wang, Xiao Xiao Lily Chen, Investigation; Olivia S Rissland, Conceptualization, Formal analysis, Supervision, Funding acquisition, Validation, Investigation, Visualization, Methodology, Writing - original draft, Project administration, Writing - review and editing

### Author ORCIDs

Michael Zavortink https://orcid.org/0000-0002-7967-1506
Lauren N Rutt https://orcid.org/0000-0001-9861-207X
Svetlana Dzitoyeva https://orcid.org/0000-0002-5471-600X
Jesslyn C Henriksen https://orcid.org/0000-0002-7371-2346
Chloe Barrington https://orcid.org/0000-0002-3884-102X
Danielle Y Bilodeau https://orcid.org/0000-0002-8086-4233
Miranda Wang https://orcid.org/0000-0002-8163-4341
Xiao Xiao Lily Chen https://orcid.org/0000-0003-1529-0568
Olivia S Rissland https://orcid.org/0000-0002-2619-6019

### Decision letter and Author response
Decision letter https://doi.org/10.7554/eLife.53889.sa1
Author response https://doi.org/10.7554/eLife.53889.sa2

## Additional files

### Supplementary files
• Transparent reporting form

### Data availability

Sequencing data have been deposited in GEO under accession code GSE140436. All data generated during this study are included in the manuscript and supporting files. Source data files have been provided for Figures 2 and 4.

The following dataset was generated:

| Author(s) | Year | Dataset title | Dataset URL | Database and Identifier |
|-----------|------|---------------|-------------|-------------------------|
| Zavortink M, Rutt LN, Dzitoyeva S, Barrington C, Bilodeau DY, Chen XL, Wang M, Rissland OS | 2019 | Me31B interaction with Kondo in stage 14 oocytes | https://www.ncbi.nlm.nih.gov/geo/query/acc.cgi?acc=GSE140436 | NCBI Gene Expression Omnibus, GSE140436 |

The following previously published datasets were used:

| Author(s) | Year | Dataset title | Dataset URL | Database and Identifier |
|---|---|---|---|---|
| Eichhorn SW, Subtelny AO, Kronja I, Orr-Weaver TL, Bartel DP | 2016 | mRNA Poly(A)-tail Changes Specified by Deadenylation Broadly Reshape Translation in Drosophila Oocytes and Early Embryos | https://www.ncbi.nlm.nih.gov/geo/query/acc.cgi?acc=GSE83616 | NCBI Gene Expression Omnibus, GSE83616 |
| Wang M, Ly M, Lugowski A, Rissland OS | 2017 | ME31B/DDX6 globally represses maternal mRNAs by two distinct mechanisms during the Drosophila maternal-to-zygotic transition | https://www.ncbi.nlm.nih.gov/geo/query/acc.cgi?acc=GSE98106 | NCBI Gene Expression Omnibus, GSE98106 |

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
