## [Decision Letter]

**Acceptance summary:**

The controlled degradation of maternally deposited mRNAs during embryogenesis has been widely studied in a variety of organisms, but the corresponding degradation of maternally deposited proteins remains uncharacterized and mysterious. By identifying the likely components of the ubiquitin system involved in the degradation of three proteins whose activity must be eliminated to proceed with development, this paper both provides a glimpse into how the system is organized and provides leverage for future efforts to tidy up our understanding of this essential step in the transition from maternal to zygotic control of embryogenesis.

**Decision letter after peer review:**

Thank you for submitting your article "Egg activation triggers clearance of maternally deposited RNA binding proteins" for consideration by *eLife*. Your article has been reviewed by two peer reviewers, and the evaluation has been overseen by Michael Eisen as the Senior and Reviewing Editor. The reviewers have opted to remain anonymous.

It is the consensus of the reviewers that this paper is not quite ready for *eLife* and they have some suggestions for modifying the manuscript that they feel might get it there.

Although it is our custom to produce a consensus review, in this case I think the separate reviews are more useful and I am leaving them as submitted. The reviewers arrived at the following during discussion that I think sums up their suggestions for how to reformulate the paper. Again, take this for what it is – suggestions from thoughtful colleagues – rather than a proscription for how you should present your work.

Overall, the reviews feel the paper should be refocused and some of the conclusions tones down or marked clearly as speculation. Those are noted in either or both reviews – the “timer” for example, and others. The paper could be refocused on the idea the ubiquitin/proteasome system is revved up at egg activation, under control of PNG, by inducing translation of Kondo. This could start by reporting on the analysis of the Bartel/Orr-Weaver data indicated that Kondo was among the strongest regulated genes. They then confirmed this on Westerns. Then, to see if it had an actual effect, they tested it on a target known to be degraded in response to PNG – mei31B (well, mei31-GFP in this case; here, they can introduce it simply as an assay) – and discovered that indeed Kondo is needed for its degradation. [They'd need to remove the rest of the RNAi screen (which is good, because that was confusing or unconvincing in a variety of ways noted in reviews) as well as the parts of the paper that try to tightly connect all this to mei41B-Cup-Tral-based RNA degradation (but over-speculate) and the speculation that PNG sets off the MZT by activating Kondo translation to then degrade mei31B which then…. (etc.). ]

Reviewer #1:

This is a well-written and clear manuscript that builds on previously published work from the Rissland lab in which they showed that the three proteins involved in translational repression, ME31B, Tral, and Cup, are degraded in a PNG-dependent manner in the early *Drosophila* embryo. Here they show that this degradation requires an E2 ubiquitin ligase, which they name Kondo, and the CTLH E3 ligase known to function with Kondo orthologs in other systems. They further demonstrate that translation of Kondo is regulated by the PNG kinase. While the identification of Kondo and the regulation of its translation by PNG are interesting, the data presented do not broadly expand the understanding of the functions that control the oocyte-to-embryo transition. However, these observations suggest novel areas for future investigation.

1) Prior work from the Rissland lab and others previously showed that ME31B, Tral, and Cup were degraded following fertilization in a PNG-dependent manner. As such, the data in Figure 1 confirm the utility of the imaging system in identifying genes that regulate ME31B stability, but do not present new information. The data in Figure 2 suggest that ME31B is degraded by the proteasome, but the limitations of the experimental system (RNAi) do not enable strong conclusions based on these data. In fact, the title for Figure 2 is misleading as the data presented to not allow one to conclude that ME31B is degraded by the ubiquitin-proteosome pathway. In addition, much of the data presented in Figure 6 were previously generated. Indeed, explicit mention of the citations for the data used to generate panels B-I need to be made. In this figure, it appears that the only data generated explicitly for his manuscript is the ME31B binding in oocytes. If this is not the case, then additional methods need to be added for ribosome profiling and poly-A tail length measurements.

2) It remains unclear what the phenotypic effects of loss of Kondo are in the embryo. Some description needs to be added. Given that Kondo mRNA appears to be expressed throughout development, the specific role of Kondo in the early embryo is challenging to reconcile. In Wang et al., the lab demonstrated that ME31B is expressed in S2 cells. It would be important to test if Kondo protein (along with the mRNA) is expressed in these cells and if so, if it is functioning to destabilize ME31B.

3) Given that ME31B binding to the Kondo mRNA is abrogated in PNG kinase mutants, a simple model might be that phosphorylation of ME31B (and associated subunits) by PNG alleviates binding to mRNA, enabling translation of many targets, including Kondo. As the authors mention, PNG- regulated translation is common to many mRNAs in the early embryo. Thus, degradation of ME31B may reinforce this translational effect, but may not be required for the change in gene expression. This would be important to test.

4) Some of the experiments would benefit from increased detail and/or quantification.

a) The RNAi screen is cleverly based on fluorescence. However, the details shown in Supplemental Table 1 do not describe how stabilization was scored. Was there a quantitative measure of fluorescence over time? If so, this should be described. If not, this seems something that should have been developed. This would help explain whether there is a biological significance to the variability in fluorescence seen in different experiments. For example, in Figure 1 the fluorescence at the zero timepoint in the plu and gnu knockdowns appears more intense than the included controls. In addition, the fluorescence in controls varies both within Figure 1 and between figures.

b) There is no information on how ubiquitin on ME31B was identified by mass spectrometry nor any discussion of these data other than in subsection “ME31B is degraded by the ubiquitin-proteasome system” where it says "see below". In addition, the methods describing the mass spectrometry performed are essentially non-existent. These methods need to be included.

c) Additional controls should be used for the pulldowns in Figure 2B and 5B. In this experimental design the authors should perform anti-GFP pulldowns on extract from flies that do not express ME31B-GFP. This is an important control.

Reviewer #2:

Zavortink et al. examine the cause of degradation of three RNA binding proteins, Mei31B, Tral, and Cup, in the maternal to zygotic transition (MZT) of *Drosophila*. In a complex these proteins block translational initiation of maternal mRNAs. They are degraded at the MZT, and remaining Mei31B degrades maternal mRNAs. These authors previously reported that destruction of these proteins depends on PNG kinase. Here they confirm this for Mei31B using a GFP fusion. They then show that ubiquitin machinery is involved in degradation of this fusion protein, and identify an E2, Kondo, and E3, CTLH, as required for this. Based on data mining and new experiments they report that Kondo translation depends on PNG. These items above are new, except for the PNG/Mei31B degradation link. The authors interpret their data into a model that PNG's role in the destruction of the Mei31B-Tral-Cup complex is in activating Kondo translation rather than due to PNG's phosphorylation of the complex's proteins.

The above conclusions are interesting, but many of the supporting conclusions are more conjectural: consistent with the model but not fully certain of having ruled out all other explanations. For example, many conclusions rely on knockdowns. With the exception of Kondo, where strong knockdown was documented, no information was provided about extent of knockdown, or the testing of multiple independent lines to rule out off-target effects. It was hard to know how to interpret experiments in which only weak knockdown can be used, such as the core barrel proteasome experiments. A partial effect, as seen, is consistent with but not proof of the hypothesis. Similar problems apply to interpretation of some of the CTLH subunit experiments. Not seeing a phenotype for CG7611 could be because of incomplete knockdown, and lack of its presence in the complexes could be a separate unrelated issue of mass spectrometry sensitivity. Lack of a phenotype for Souji knockdown in terms of RanBPM-Mei31B can also not be interpreted mechanistically, although such an interpretation is implied.

Other cases of imprecision or conjecture include the following situations that go beyond wording problems, and require experimentation to resolve:

In subsection “The E2 conjugating enzyme UBC-E2H is required for the degradation of ME31B, TRAL, and Cup”, the authors state that both wild-type and fusion proteins were completely stabilized when UBC-E2H was depleted. But Figure 3C shows this only for the fusion protein. In the same figure the wildtype protein's levels decrease with time. This is a potentially important difference that needs attention.

The paper posits that a timer is set by egg activation through the agency of PNG to cause destruction of Mei31B at the MZT. This timing model is not tested. It could simply be that MZT can only happen after egg activation, and that egg activation, and PNG, is necessary for subsequent clearance of these proteins without serving as a timer.

Several times the paper says "and likely Tral and Cup", without testing it. This weakens the conclusions.

The paper concludes that because CTLH binds Mei31B in the absence of PNG, PNG phosphorylation must not trigger Mei31B destruction. This is consistent with the authors' conclusion but do not prove it. For that, a non-phosphorylatable Mei31B must be tested for ubiquitin-mediated destruction.

Other phenomena that may contribute are not considered. For example Wispy also regulates poly A tail length. And although SUMO modification, and Dhd, are noted they are not considered in the model.

Some interpretations are overstatements. For example, the authors state that ubiquitnation of ME31B is "completely absent" in Kondo knockdown, but they can only conclude that any ubiquitination is beyond their detection limit. Conclusions about composition of active CTLH also go well beyond the data

[Editors' note: further revisions were suggested prior to acceptance, as described below.]

I am pleased to inform you that your article, "Egg activation triggers clearance of maternally deposited RNA binding proteins", has been provisionally accepted for publication in *eLife*.

As you can see from the comments from the reviewers, there is disagreement about whether the additional data addresses the concerns raised. However, under our COVID publication policies, we recognize that it is not practical to address them at this time. We therefore ask, as a final step prior to publication, that you do your best to address these limitations in the Discussion.

Reviewer #1:

This manuscript builds on previously published work from the Rissland lab in which they showed that the three proteins involved in translational repression, ME31B, Tral, and Cup, are degraded in a PNG-dependent manner in the early *Drosophila* embryo. In this revision, Zavortink et al. now provide quantification of their florescent imaging and structural modelling of the *Drosophila* CTLH complex. However, due to the current limits on experimentation very few of the prior concerns have been addressed experimentally. The prior submission was "not quite ready for *eLife*", and because of the limited new data this manuscript does not broadly expand the understanding of the functions that control the oocyte-to-embryo transition.

The additional controls strengthen the conclusions drawn in the manuscript. Nonetheless, previous reviews highlighted a number of avenues for experimentation that would broaden the scope of the manuscript. These included phenotypic characterization of Kondo loss-of-function that could help explain why ME31B is stable in some cells and not others and investigation of whether ME31B phosphorylation by PNG alleviates binding to Kondo mRNA. The authors argue that these are beyond the scope of the current manuscript, but without some additional data the current manuscript does not significantly change the field.

As it stands, the authors argue that translation of Kondo is triggered by PNG and that Kondo expression is necessary for ME31B expression. (Indeed, this is the model in Figure 7.) But, clearly this cannot be the full story since Kondo is expressed in cells where ME31B is stable. Furthermore, ME31B binds Kondo mRNA (and this is completely missing from the over-simplified model). Thus, it is likely that there are other changes triggered by PNG-mediated phosphorylation that result in ME31B degradation apart from Kondo expression. The identification of Kondo, while interesting, does not broadly shift the understanding of the previously identified PNG-dependent degradation of ME31B, TRAL and Cup.

The only additional figure in the resubmitted manuscript are the inclusion of the structural modelling of the *Drosophila* CTLH complex. The impact of these data on the paper (which is written to address protein dynamics during the MZT) are unclear. This modelling, while potentially informative, does not provide an explanation for the difference between the two RanBPM isoforms or how this structure impacts the triggered degradation of ME31B during the MZT.

While the authors may argue that current research shutdowns result in the limited new experiments that were added on resubmission, this would not preclude substantive textual changes as suggested in the prior review. Thus, it was surprising that the paper is largely structured as it was previously. In addition, the Materials and methods continue to lack clarity. Just because mass spectrometry was performed at a fee for service facility does not mean that an explanation of how the peptides were identified and quantified can be excluded from the methods of a paper. The methods in the Wang et al. reference has a similarly vague explanation and thus does not provide any additional information. This is important information for future researchers looking to build or extend this work. Similarly, the information on the lysis conditions for the various IPs is unclear. There is a section on "Isolation of embryos" that explains a single lysis condition. Were these the conditions used for the coIP experiments? This is important for anyone wanting to repeat or extend these figures. Along these lines, it would be useful to explain the surprising background precipitation seen in Figures 5B and 6B (for ME31B and ME31B-GFP) and Figure 6C (for Cup) in the IgG control. These non-specific pulldowns of the target proteins suggest lysis or IP conditions might not be very stringent.

Reviewer #2:

I thank the authors for considering my comments and for addressing them as well as possible under COVID-19 constraints. Softening the language where experiments were not possible is acceptable under current situation, and including the Cao et al. study and the implications of the Qiao et al. study addressed some of my major concerns. The authors could consider adding brief acknowledgement of limitations such as not knowing extent of depletion in the knockdowns, noting that it was not possible to test this at present.

The one comment that was not fully addressed was my Minor Comment #1. The authors' response does address my concern about whether Mei31B-GFP is functional. But they still retain imprecise language in the text, using "Mei31B" when what they tested was Mei31B-GFP, or in statements like "We set out to identify E2 conjugating enzymes and E3 ligases responsible for the degradation of ME31B by.… RNAi screen. As before, we monitored ME31B decay by GFP fluorescence", or re-state clauses like "knockdown of UBC-E2H… blocked degradation of Mei31B…", when degradation of Mei31B-GFP was what was monitored in the GFP fluorescence assay. This makes it confusing for readers and is not fully accurate.

---

## [Author Response]

Overall, the reviews feel the paper should be refocused and some of the conclusions tones down or marked clearly as speculation. Those are noted in either or both reviews – the “timer” for example, and others. The paper could be refocused on the idea the ubiquitin/proteasome system is revved up at egg activation, under control of PNG, by inducing translation of Kondo. This could start by reporting on the analysis of the Bartel/Orr-Weaver data indicated that Kondo was among the strongest regulated genes. They then confirmed this on Westerns. Then, to see if it had an actual effect, they tested it on a target known to be degraded in response to PNG – mei31B (well, mei31-GFP in this case; here, they can introduce it simply as an assay) – and discovered that indeed Kondo is needed for its degradation. [They'd need to remove the rest of the RNAi screen (which is good, because that was confusing or unconvincing in a variety of ways noted in reviews) as well as the parts of the paper that try to tightly connect all this to mei41B-Cup-Tral-based RNA degradation (but over-speculate) and the speculation that PNG sets off the MZT by activating Kondo translation to then degrade mei31B which then…. (etc.). ]

We thank the reviewers for their thoughtful comments about our manuscript. We especially appreciated their comments that our manuscript is “well-written and clear,” “interesting,” and suggest “novel areas for future investigation.”

Because our lab was shut down due to COVID-19, we were unable to perform all of the suggested experiments, especially those experiments investigating the impact of Kondo-mediated degradation. However, we have extended our results into the overall organization of the *Drosophila* CTLH E3 complex and made the observation that this complex differs from that in yeast (and likely in humans), especially in terms of substrate recognition. We believe these additional, homology-based structural results take our manuscript one step further and increase its relevance for scientists in the protein degradation field.

Reviewer #1:1) Prior work from the Rissland lab and others previously showed that ME31B, Tral, and Cup were degraded following fertilization in a PNG-dependent manner. As such, the data in Figure 1 confirm the utility of the imaging system in identifying genes that regulate ME31B stability, but do not present new information.

In our previous work (Wang et al., 2017), we had only investigated the stability of ME31B, TRAL, and Cup in *png50* embryos, and thus had not shown the involvement of GNU or PLU. Although this result in Figure 1 was unsurprising (given that the three factors act in a complex), it was important. This experiment also demonstrated the utility of our approach. In addition, we had not previously tested the importance of fertilization, as we do in Figure 1E. Together, this figure demonstrates the importance of egg activation and the PNG kinase, rather than fertilization, for the destruction of ME31B.

The data in Figure 2 suggest that ME31B is degraded by the proteasome, but the limitations of the experimental system (RNAi) do not enable strong conclusions based on these data. In fact, the title for Figure 2 is misleading as the data presented to not allow one to conclude that ME31B is degraded by the ubiquitin-proteosome pathway.

We agree that the RNAi system has limitations (e.g., for many of our knockdowns, we have been unable to determine the degree of depletion at the protein level), and so we have changed the figure and its discussion in our revised manuscript in several ways. First, we have now quantified the fluorescence in Figure 2C (see Figure 2D). Based on this quantitation, we conclude that degradation of ME31B-GFP is slowed in *Rpn10-*depleted embryos.

Second, due to COVID-19, we were unable to perform more direct experiments (such as injected MG132 into embryos). However, a contemporaneous study from the Lipshitz and Wahle labs performed this experiment and showed that MG132 treatment stabilized ME31B (Cao et al., bioRxiv, 2019). We now cite their result in our paper, and suggest that, together, these data strongly suggest that ME31B is degraded by the ubiquitin-proteasome pathway.

In addition, much of the data presented in Figure 6 were previously generated. Indeed, explicit mention of the citations for the data used to generate panels B-I need to be made. In this figure, it appears that the only data generated explicitly for his manuscript is the ME31B binding in oocytes. If this is not the case, then additional methods need to be added for ribosome profiling and poly-A tail length measurements.

The reviewer is correct that the ribosome profiling and poly(A)-tail length measurements were previously performed as transcriptome-wide studies in Eichhorn et al., 2016. We apologize for the confusion and have now made this clear in the Figure legend.

2) It remains unclear what the phenotypic effects of loss of Kondo are in the embryo. Some description needs to be added. Given that Kondo mRNA appears to be expressed throughout development, the specific role of Kondo in the early embryo is challenging to reconcile. In Wang et al., the lab demonstrated that ME31B is expressed in S2 cells. It would be important to test if Kondo protein (along with the mRNA) is expressed in these cells and if so, if it is functioning to destabilize ME31B.

We had only performed a few preliminary experiments into the issues raised by the reviewer before our campus was shut-down due to COVID-19. As noted in other papers (Kronja et al., 2014), embryos depleted of Muskelin do produce viable flies, and we have also seen that embryos depleted of Kondo produce adults. However, we have not performed a careful phenotypic investigation, and so we have not commented on the phenotype in our current manuscript.

It is worth noting that it does appear that some Kondo alleles are recessive-lethal, which is consistent with Kondo having additional roles outside the embryo. Based on this reported phenotype, fairly ubiquitous Kondo expression, and what is known in other organisms, we suspect that Kondo may also lead to the degradation of other proteins through the CTLH complex. In yeast, the CTLH complex has several different substrate adaptor proteins (e.g., Gid4 and Gid10) that in turn target different proteins. We suspect that something similar might be at work in *Drosophila*, but identifying the recognized degron and the role of Kondo outside of the embryo extends beyond the scope of the current manuscript.

Unfortunately, S2 cells do not recapitulate key aspects of the embryo system (e.g. ME31B is very stable in these cells [t_1/2_ ~6 hours]), which precludes use of S2 cells for these experiments. We cannot speak to the critical differences between the two systems, especially given the strong likelihood that we have not identified all components of the CTLH complex. Exploring the requirements for ME31B degradation and how the protein is recognized is an exciting direction for future.

3) Given that ME31B binding to the Kondo mRNA is abrogated in PNG kinase mutants, a simple model might be that phosphorylation of ME31B (and associated subunits) by PNG alleviates binding to mRNA, enabling translation of many targets, including Kondo. As the authors mention, PNG- regulated translation is common to many mRNAs in the early embryo. Thus, degradation of ME31B may reinforce this translational effect, but may not be required for the change in gene expression. This would be important to test.

We agree with the reviewer that this idea is very tempting, and would be consistent with the viability of Muskelin- or Kondo-depleted embryos. However, because the focus of our manuscript is on how ME31B is degraded, we believe that this issue falls outside the scope of our current study, but will be exciting to investigate in the future.

4) Some of the experiments would benefit from increased detail and/or quantification.a) The RNAi screen is cleverly based on fluorescence. However, the details shown in Supplemental Table 1 do not describe how stabilization was scored. Was there a quantitative measure of fluorescence over time? If so, this should be described. If not, this seems something that should have been developed.

We thank the reviewer for this suggestion and have now included quantification for Figure 1A and 2C.

In our original screen, because we were primarily interested in identifying genes whose depletion recapitulated the *png50* phenotype, we scored phenotypes based on qualitative measurements of fluorescence, as we now make clear in the Materials and methods section. Because almost the entirety of the screen is now not described (although still available as Figure 2—source data 1), we have left these scores as qualitative.

This would help explain whether there is a biological significance to the variability in fluorescence seen in different experiments. For example, in Figure 1 the fluorescence at the zero timepoint in the plu and gnu knockdowns appears more intense than the included controls. In addition, the fluorescence in controls varies both within Figure 1 and between figures.

We thank the reviewer for making this point. Much of the variation between wild-type embryos, we suspect is due to differences in embryo age that occurs during the initial 1-hour collection and preparation prior to placing the embryos on the slide. In order to control for this variation in our quantitation, we normalized the fluorescence to the intensity in the first time point. Similarly, we suspect variation between controls in different figures is due to day-to-day differences in collections, dechorionation times, etc. Of course, this variation is one of the reasons why we included an mCherry in each experiment and in the screen itself.

In *png50* mutants or other lines lacking key degradation factors, the fluorescence is more homogenous and more intense because there is limited degradation that occurs during the collection stage. We now explain this point in the main text.

b) There is no information on how ubiquitin on ME31B was identified by mass spectrometry nor any discussion of these data other than in subsection “ME31B is degraded by the ubiquitin-proteasome system” where it says "see below". In addition, the methods describing the mass spectrometry performed are essentially non-existent. These methods need to be included.

The mass spectrometry was performed as fee-for-service in the SickKids core, as is now reflected in the Materials and methods section. The identification of ubiquitin peptides was performed manually because ubiquitin is produced as fusion proteins with ribosomal proteins; the peptides mapping to ubiquitin were manually identified.

c) Additional controls should be used for the pulldowns in Figure 2B and 5B. In this experimental design the authors should perform anti-GFP pulldowns on extract from flies that do not express ME31B-GFP. This is an important control.

We thank the reviewer for this feedback. We have now included several control experiments, which substantially strengthen our manuscript. In Figure 4F, we compared anti-GFP immunoprecipitations from *w1118* and ME31B-GFP embryos. As expected, we detect ME31B-GFP and RanBPM only in the ME31B-GFP embryos. Moreover, in Figure 6C, we immunoprecipitated Cup using antibodies recognizing the endogenous protein from both *w1118* and *png50* embryos. Consistent with our ME31B-GFP immunoprecipitations, we were able to immunoprecipitate the long isoform of RanBPM in both lysates.

Reviewer #2:[…] The above conclusions are interesting, but many of the supporting conclusions are more conjectural: consistent with the model but not fully certain of having ruled out all other explanations. For example, many conclusions rely on knockdowns. With the exception of Kondo, where strong knockdown was documented, no information was provided about extent of knockdown, or the testing of multiple independent lines to rule out off-target effects.

We agree with the reviewer that, ideally, the extent of depletion would be known for key factors and multiple independent lines would be used for our knock-downs. We have made use all available reagents that we can find: e.g., we use an antibody recognizing RanBPM to determine its knockdown and, whenever possible, have used multiple RNAi lines. Unfortunately, due to COVID-19, we were not able to fully explore alternative lines, and have changed the text accordingly.

It was hard to know how to interpret experiments in which only weak knockdown can be used, such as the core barrel proteasome experiments. A partial effect, as seen, is consistent with but not proof of the hypothesis.

As noted in our response to reviewer 1, due to COVID-19, we were unable to perform more direct experiments (such as injected MG132 into embryos). However, a contemporaneous study from the Lipshitz and Wahle labs performed this experiment and showed that MG132 treatment stabilized ME31B (Cao et al., bioRxiv, 2019). We now cite their result in our paper, and suggest that, together, these data strongly suggest that ME31B is degraded by the ubiquitin-proteasome pathway.

Similar problems apply to interpretation of some of the CTLH subunit experiments. Not seeing a phenotype for CG7611 could be because of incomplete knockdown, and lack of its presence in the complexes could be a separate unrelated issue of mass spectrometry sensitivity.

Unfortunately, making mutants for putative CTLH components and Kondo has been challenging (perhaps because these proteins have functions outside embryogenesis), and, with COVID-10, we have had to rely on RNAi lines. The reviewer is correct that we cannot exclude the possibility that RNAi is incomplete (which we cannot currently test because we lack antibodies for CG7611) and unrelated mass spectrometry issues. We have softened the language around our discussion of CG7611.

Lack of a phenotype for Souji knockdown in terms of RanBPM-Mei31B can also not be interpreted mechanistically, although such an interpretation is implied.

A new cryo-EM structure of the yeast CTLH complex (known as the Gid complex) was published in February that provided substantial insight into the likely overall organization of the *Drosophila* complex through homology (Qiao et al., 2020). RanBPM is part of a scaffold module that links to the substrate recognition complex, while Sou and Kaz are part of the catalytic domain, linked to the rest of the complex through Hou.

Using homology-based structure modeling, we find highly similar predicted structural organizations for RanBPM/Gid1, Hou/Gid5, Kaz/Gid2, and Sou/Gid9, strongly suggesting that the *Drosophila* complex is organized the same way. In this respect, it is unsurprising that knock-down of Sou did not affect the ME31B-RanBPM interaction.

One surprise did emerge from our analysis, which we know describe in the main text and Figure 5. In the yeast complex, Gid4 (which is linked to the rest of the complex predominantly through Gid5) recognizes substrate proteins. We are unable to identify a putative *Dm* Gid4 or Gid5, using either yeast or human primary protein sequences or structure-based reverse searches based on the yeast structure. Indeed, closer inspection of RanBPM and Hou threaded through the yeast structure revealed striking differences at their predicted interface with Gid5/4: a loop in Gid1 that interacts with Gid4 is missing in RanBPM; a long C-terminal expansion in Gid8 that wraps around Gid5 is also missing in Hou. Together, these results raise the possibility that the *Drosophila* complex has a different organization than the yeast one, which will be exciting to investigate in the future.

Other cases of imprecision or conjecture include the following situations that go beyond wording problems, and require experimentation to resolve:In subsection “The E2 conjugating enzyme UBC-E2H is required for the degradation of ME31B, TRAL, and Cup”, the authors state that both wild-type and fusion proteins were completely stabilized when UBC-E2H was depleted. But Figure 3C shows this only for the fusion protein. In the same figure the wildtype protein's levels decrease with time. This is a potentially important difference that needs attention.

We have previously seen that ME31B-GFP is slightly more stable than the endogenous protein (Wang et al., 2017) although the underlying mechanism will likely require a better understand of how ME31B is recognized by the E3. We have clarified this point in the text.

The paper posits that a timer is set by egg activation through the agency of PNG to cause destruction of Mei31B at the MZT. This timing model is not tested. It could simply be that MZT can only happen after egg activation, and that egg activation, and PNG, is necessary for subsequent clearance of these proteins without serving as a timer.

We have removed all mention of the timer from the text.

Several times the paper says "and likely Tral and Cup", without testing it. This weakens the conclusions.

For technical reasons, we cannot directly investigate some aspects of TRAL and Cup biology (e.g., interactions with the CTLH complex or ubiquitination). Nonetheless, as described above, we now include immunoprecipitations against endogenous Cup and observe interactions with RanBPM in wild-type and *png50* embryos.

The paper concludes that because CTLH binds Mei31B in the absence of PNG, PNG phosphorylation must not trigger Mei31B destruction. This is consistent with the authors' conclusion but do not prove it. For that, a non-phosphorylatable Mei31B must be tested for ubiquitin-mediated destruction.

We agree with the reviewer that, in an ideal scenario, this experiment would be important to do. However, it is not feasible for several reasons. First, ME31B is phosphorylated by PNG but the sites are currently unknown (Hara et al., 2018). Second, it remains unclear if ME31B directly recognized or if Cup or TRAL are also recognized. Each of these two proteins is directly phosphorylated by PNG (Hara et al., 2018). Nonetheless, it is clear that both ME31B and Cup can interact with the E3 ligase in the absence of PNG activity, which strongly suggests that the phosphorylation is not required for their recognition by the E3 ligase.

Other phenomena that may contribute are not considered. For example Wispy also regulates poly A tail length. And although SUMO modification, and Dhd, are noted they are not considered in the model.

We agree that other phenomena may contribute to the regulation of *Kondo* translation. Indeed, Wispy does regulate Kondo poly(A)-tail length, as determined from data from Eichhorn et al., 2016. Interestingly, this paper also observed that Wispy does not affect the relative translation of mRNAs, and so understanding the role of Wispy for *Kondo* translation will require more substantial investigation. Similarly, although Sumo regulation appears involved in the degradation of ME31B, understanding this involvement falls outside the scope of our current study. (Note that because our discussion of our RNAi screen has been simplified, we have now removed reference to Sumo regulation in the main text, although the source data remains.)

Some interpretations are overstatements. For example, the authors state that ubiquitnation of ME31B is "completely absent" in Kondo knockdown, but they can only conclude that any ubiquitination is beyond their detection limit.

We have changed the text in multiple places to reflect this point.

Conclusions about composition of active CTLH also go well beyond the data.

As noted above, we have performed an in-depth analysis of the CTLH complex, modeling on the *Sc.* Gid complex. We have also modified the text to soften our statements, where pointed by the reviewer.

[Editors' note: further revisions were suggested prior to acceptance, as described below.]

Reviewer #1:[…]As it stands, the authors argue that translation of Kondo is triggered by PNG and that Kondo expression is necessary for ME31B expression. (Indeed, this is the model in Figure 7.) But, clearly this cannot be the full story since Kondo is expressed in cells where ME31B is stable. Furthermore, ME31B binds Kondo mRNA (and this is completely missing from the over-simplified model).

We have removed the model from Figure 7.

Thus, it is likely that there are other changes triggered by PNG-mediated phosphorylation that result in ME31B degradation apart from Kondo expression. The identification of Kondo, while interesting, does not broadly shift the understanding of the previously identified PNG-dependent degradation of ME31B, TRAL and Cup.

We agree with the reviewer that other changes due to PNG activity may be important for ME31B degradation and state as much in our discussion. However, because RanBPM and ME31B/Cup interact in PNG mutant embryos, the conclusion stands that PNG phosphorylation is not required for the CTLH complex to recognize ME31B.

We have added an additional point to the Discussion that may shed some light as to why ME31B is stable outside of the MZT: according to RNA-seq published on FlyBase, Muskelin expression is highly restricted to the ovary and early embryo, and shows little expression in standard cell lines, such as S2 cells. Because Muskelin is required for ME31B degradation, the tissue specificity of Muskelin, together with the translational activation of *Kondo*, may provide an explanation as to why ME31B, TRAL, and Cup are degraded specifically in the early embryo (although there may be additional layers of regulation as well). Another (non-mutually exclusive) possibility is that Cup is the major substrate for the CTLH complex and that ME31B and TRAL are only degraded when Cup is also present. Like Muskelin, Cup expression is highly restricted, and it is only robustly expressed in the ovary and testes (with some minor expression in the imaginal disc). Based on our current understanding, we cannot exclude these different possibilities, but we agree with the reviewer that these are important issues to understand in the future.

In addition, the Materials and methods continue to lack clarity. Just because mass spectrometry was performed at a fee for service facility does not mean that an explanation of how the peptides were identified and quantified can be excluded from the methods of a paper. The methods in the Wang et al. reference has a similarly vague explanation and thus does not provide any additional information. This is important information for future researchers looking to build or extend this work.

We have included more information about the mass spectrometry in our Materials and methods section.

Similarly, the information on the lysis conditions for the various IPs is unclear. There is a section on "Isolation of embryos" that explains a single lysis condition. Were these the conditions used for the coIP experiments? This is important for anyone wanting to repeat or extend these figures.

We apologize for the confusion and have tried to clarify our methods section. Briefly, the embryo isolates are made the same way for all experiments and stored at -80C. Depending on the nature of the experiment, we then modify the buffer used to dilute the sample or to wash the beads.

Along these lines, it would be useful to explain the surprising background precipitation seen in Figures 5B and 6B (for ME31B and ME31B-GFP) and Figure 6C (for Cup) in the IgG control. These non-specific pulldowns of the target proteins suggest lysis or IP conditions might not be very stringent.

ME31B, Cup, and other RNA binding proteins can be sticky and bind beads non-specifically, and we have occasionally seen low levels of the proteins in our controls even with stringent buffers. Nonetheless, the interaction with RanBPM is specific in 5B, 6B, and 6C. Note that in our standard conditions beads are washed three times and transferred to a new tube (as we describe in our Materials and methods section). That being said, we do find that the interaction between RanBPM and ME31B is weaker than and harsher buffers, such as RIPA, disrupt the interaction.

Reviewer #2:I thank the authors for considering my comments and for addressing them as well as possible under COVID-19 constraints. Softening the language where experiments were not possible is acceptable under current situation, and including the Cao et al. study and the implications of the Qiao et al. study addressed some of my major concerns. The authors could consider adding brief acknowledgement of limitations such as not knowing extent of depletion in the knockdowns, noting that it was not possible to test this at present.

We have included this suggestion in discussing the results from our screen.

The one comment that was not fully addressed was my Minor Comment #1. The authors' response does address my concern about whether Mei31B-GFP is functional. But they still retain imprecise language in the text, using "Mei31B" when what they tested was Mei31B-GFP, or in statements like "We set out to identify E2 conjugating enzymes and E3 ligases responsible for the degradation of ME31B by.… RNAi screen. As before, we monitored ME31B decay by GFP fluorescence", or re-state clauses like "knockdown of UBC-E2H… blocked degradation of Mei31B…", when degradation of Mei31B-GFP was what was monitored in the GFP fluorescence assay. This makes it confusing for readers and is not fully accurate.

We apologize for the confusion and have fixed this.